# CLIP-FMoE: Scalable CLIP via Fused Mixture-of-Experts with Enforced Specialization

**Luong Tran**[1]    **Lan-Cuong Nguyen**[1]    **Huynh Dang Nguyen**[1]    **Dat Nguyen Cong**[1]
**Dung D. Le**[2]    **Van Nguyen**[1]*
[1]FPT Software AI Center
[2]Center for AI Research, VinUniversity

## Abstract

Mixture-of-Experts (MoE) architectures have emerged as a promising approach for scaling deep learning models while maintaining computational efficiency. However, existing MoE adaptations for Contrastive Language-Image Pre-training (CLIP) models suffer from significant computational overhead during sequential training and degradation of zero-shot capabilities. To address these limitations, we propose CLIP-FMoE, a novel approach that integrates MoE architecture into CLIP fine-tuning. Our method uses Isolated Constrained Contrastive Learning, a pipeline that trains specialized experts on cluster-based data partitions to accelerate expert specialization. Additionally, we introduce a Fusion Gate mechanism to mitigate catastrophic forgetting of pre-trained knowledge. Extensive experiments across multiple benchmarks demonstrate that our approach achieves consistent improvements on downstream tasks while preserving zero-shot capabilities. Furthermore, our method demonstrates robust performance across varying context lengths, making it particularly suitable for diverse real-world applications.

## 1 Introduction

Contrastive vision–language models, most notably CLIP (Radford et al., 2021), demonstrate strong zero-shot generalization across diverse tasks. A natural approach to enhancing CLIP is to scale up the model size, as demonstrated by follow-up studies such as EVA-CLIP (Fang et al., 2023), which show that larger variants consistently outperform their smaller counterparts. However, pure parameter scaling comes at a steep cost: keeping large models stable requires much longer training schedules, significantly larger batch sizes, and sprawling GPU clusters, turning it into a highly resource-intensive effort that soon outstrips the means of most practitioners.

These scaling challenges motivate the need for alternative approaches that can enhance CLIP's capacity without the prohibitive cost of uniformly increasing model size. One promising direction is the Mixture-of-Experts (MoE) architecture (Jacobs et al., 1991), which has recently gained attention as an efficient way to scale large models sparsely. Instead of activating the entire network for each input, MoE selectively activates only a subset of "experts", enabling faster processing, improved scalability for large models, and greater computational efficiency.

In the context of vision–language models, MoE offers a compelling solution: It allows us to expand CLIP's modeling power while keeping inference efficient. Recent works such as CLIP-MoE (Zhang et al., 2024c) and CLIP-UP (Wang et al., 2025b) have demonstrated the potential of MoE to improve performance without fully retraining massive models. However, prior works do not ensure the specialization of experts in MoE architectures (i.e., avoiding trivial expert behavior) or address the scalability of the methods (i.e., the number of concurrently trained experts). This is our first primary research focus.

On the other hand, CLIP struggles with fine-grained visual features, likely because its web-crawled pre-training captions are short and noisy, often failing to capture detailed image content. To address this, researchers have turned to fine-grained long-caption datasets. Works like DreamLIP (Zheng et al., 2024) and FLAIR (Xiao et al., 2025) recaptures roughly 30M web-scraped images using

---

*Corresponding author.

multimodal large language models (LLMs), achieving strong results in fine-grained tasks, but still lagging behind CLIP in coarse-grained zero-shot classification. This gap can be attributed to differences in data volume. Alternatively, some methods fine-tune CLIP on synthetic fine-grained data. FG-CLIP (Xie et al., 2025), for example, augments CLIP-400M with 1.6B LLM-generated samples, but heavy reliance on synthetic captions risks a "garbage-in, garbage-out" cycle (as the multi-modal LLMs use vision encoders trained on web-scale datasets) and may introduce hallucination noise. Other approaches such as LongCLIP (Zhang et al., 2024a), TULIP (Najdenkoska et al., 2025), and FineCLIP (Jing et al., 2024)) train on smaller fine-grained sets and extend context lengths, but often sacrifice generalization on zero-shot benchmarks.

To mitigate catastrophic forgetting, the continual learning literature has explored a diverse set of strategies. Significant attention has been directed toward Mixture-of-Experts (MoE) and prompt-based frameworks, such as MoE-Adapters (Yu et al., 2024) and L2P (Wang et al., 2022), which dynamically adapt model architecture or input prompts. Parallel efforts have focused on regularization-based techniques, notably ZSCL (Zheng et al., 2023), to constrain weight updates, while others have employed orthogonality-based constraints like PGP (Qiao et al., 2024) to minimize interference between tasks. Additionally, feature compression methods such as LADA (Luo et al., 2025) have been proposed to enhance scalability. Distinguishing itself from these existing paradigms, our work introduces a fusion gate mechanism, designed to explicitly preserve pretrained knowledge during the learning process while allowing experts to acquire new complementary representations.

Those existing approaches either neglect the valuable pre-trained knowledge embedded in vision-language models or fail to preserve the robust Internet keyword labeling knowledge that remains unaffected by hallucination issues. Our second primary objective addresses this gap by developing a solution to mitigate catastrophic forgetting while simultaneously improving performance through training on additional fine-grained datasets.

Our main contributions are as follows.

- We introduce CLIP-FMoE, a novel approach to scale vision–language models that combines a knowledge-fusion with an Isolated Constrained Contrastive Learning (ICCL) mechanism. This design preserves and leverages pre-trained knowledge while efficiently adapting to new data under limited computational resources.

- We demonstrate that CLIP-FMoE helps ensure the specialization of experts by learning abundant, distinct knowledge, through data clustering, and reducing the effect of token imbalance routing in MoE, thanks to the train-before-unify pipeline.

- CLIP-FMoE shows consistent improvement on various downstream tasks across different settings of context length and dataset size.

## 2 RELATED WORKS

### 2.1 VISION-LANGUAGE MODELS

Vision-Language foundation models like CLIP (Radford et al., 2021), trained on large-scale datasets, show impressive generalization across multiple benchmarks. However, despite CLIP's strong zero-shot performance, its reliance on short and noisy pre-training captions limits effectiveness in specific scenarios requiring fine-grained understanding and constrains its scalability. Recent approaches to improve CLIP can be categorized into several directions. Improvements in contrastive learning focus on revisiting the original contrastive objective, particularly by improving the quality of negative samples within training batches (Ma et al., 2024). The data approach leverages datasets with improved captions generated by multi-modal large language models (MLLMs). For pre-training, works like DreamLIP (Zheng et al., 2024) and FLAIR (Xiao et al., 2025) train on 30M high-quality samples to improve detailed understanding and boost generalization. For fine-tuning, FGCLIP (Xie et al., 2025) further fine-tunes CLIP on 1.6B synthetic LLM-generated samples. Context extension methods such as LongCLIP (Zhang et al., 2024a), TULIP (Najdenkoska et al., 2025), and FineCLIP (Jing et al., 2024) focus on extending context windows to utilize high-quality datasets containing long, descriptive captions effectively. An orthogonal approach is to scale model capacity. The EVA-CLIP series grows model size from 5B parameters (EVA-02-CLIP (Fang et al., 2023)) to 18B parameters (EVA-CLIP-18B (Sun et al., 2024)), achieving 80% zero-shot top-1 accuracy across 27 benchmarks.

## 2.2 MIXTURE-OF-EXPERTS

The Mixture-of-Experts (MoE) paradigm (Jacobs et al., 1991) enables dramatic increases in model capacity without a proportional rise in computation by activating only a sparse subset of experts per token (Shazeer et al., 2017; Mu & Lin, 2025). Empirically, this sparsity mechanism has unlocked large performance gains across vision, language, and multimodal tasks (Rajbhandari et al., 2022; Dai et al., 2024; Wu et al., 2024; Mustafa et al., 2022), and enables expert specialization for improved task coverage (Shen et al., 2023; Zhu et al., 2025; Lu et al., 2024). A central bottleneck in MoE development lies in the construction and initialization of effective experts. Training an MoE from scratch can be prohibitively resource-intensive. To address this, Sparse Upcycling (Komatsuzaki et al., 2023) introduces a cost-efficient initialization scheme by cloning the MLP layers of a dense pretrained model to serve as multiple experts. However, the homogeneity of replicated MLPs reduces further specialization, limiting the upper bound of the model capabilities and leading to suboptimal performance (He et al., 2024; Jiang et al., 2025; Nakamura et al., 2025). To address this in vision-language models, CLIP-MoE (Zhang et al., 2024c) sequentially trains experts via a cluster-and-contrast scheme, while MODE (Ma et al., 2024) uses ensemble learning, training separate experts in disjoint subsets without an MoE layer.

## 2.3 MODEL MERGING

The problem of combining pre-trained models with task-specific knowledge has been explored through various paradigms. Model merging techniques such as task arithmetic (Ilharco et al., 2023) enable linear combinations of model weights to compose capabilities, while LiNeS (Wang et al., 2025a) introduces post-training layer scaling to prevent catastrophic forgetting during merging. Temporal merging approaches (Dziadzio et al., 2025) address how to combine multimodal models trained at different time points, and recent work on task arithmetic in tangent space (Ortiz-Jimenez et al., 2023) demonstrates improved editing of pre-trained models through geometric considerations. Beyond static merging, dynamic composition methods have gained attention: mixture-of-experts architectures (Fedus et al., 2022b) employ gating mechanisms to route inputs across specialized modules, while Model Soups (Zimmer et al., 2024) demonstrate that averaging weights of multiple fine-tuned models improves accuracy. Unlike prior merging methods that operate on pre-trained components, our approach learns the fusion dynamically and hierarchically throughout the training process.

## 3 PRELIMINARIES

### 3.1 CONTRASTIVE LEARNING

Contrastive learning (van den Oord et al., 2018) has become a foundational technique for aligning representations across different modalities. CLIP (Radford et al., 2021) measures the similarity of images and text pairs $\{I_i, T_i\}_{i=1}^{B}$ in a batch $B$ using a function $s(\cdot, \cdot)$, which pulls matching pairs closer in the embedding space while pushing mismatched pairs apart.

To achieve this, a symmetric InfoNCE objective is applied across the batch. For each image–text pair, the loss encourages the model to assign higher similarity to the true pairing than to all other combinations in the batch:

$$\text{InfoNCE}(I, T) = -\frac{1}{B} \sum_{i=1}^{B} \log \frac{\exp\big(s(I_i, T_i)\big)}{\sum_{j=1}^{B} \exp\big(s(I_i, T_j)\big)}. \tag{1}$$

### 3.2 MIXTURE-OF-EXPERTS

In Transformer architecture, a Mixture-of-Experts (MoE) layer replaces the conventional MLP with a collection of $N$ experts $\{E^{(i)}\}_{i=1}^{N}$ controlled by a router $R$, where each expert $E^{(i)}$ is an independent MLP. The routing process begins by computing expert assignment logits of a given input token representation $x$ through $R$ parameterized by a weight matrix $W$. The $K$ experts out of $N$ with the highest logits are then selected by TopK operation, and their corresponding logits are then normalized using the softmax function. This yields the routing probability distribution $R(x)$ over

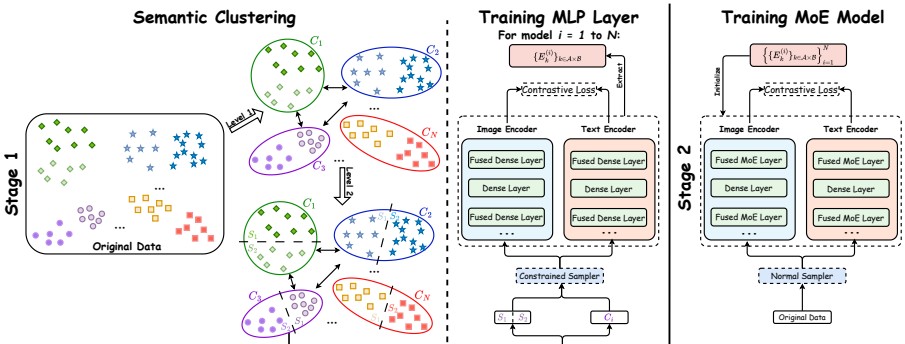

Figure 1: Illustration of our main training pipeline. Stage 1 performs semantic clustering to group original data into distinct clusters ($C_0$, $C_1$, $C_2$, ..., $C_N$) (left). Data from each cluster trains a corresponding MLP layer, with all MLPs training concurrently (middle). In Stage 2, the trained MLP weights from Stage 1 serve as initialization for the MoE experts. During this stage, only the router network is trained while the expert weights remain frozen (right).

the selected experts among all $N$ available experts:

$$E_{moe} = \sum_{i=1}^{N} R(x)_i \cdot E^{(i)}(x), \quad R(x) = \text{Softmax}(\text{TopK}(x \cdot W)), \qquad (2)$$

where $R(x)_i$ represents the $i$-th routing weight produced by the router network $W$.

This framework allows models to scale in size and capacity without proportionally increasing computation costs compared to dense counterparts. To balance token utilization across experts, Fedus et al. (2022a) proposed a regularization term. Given $N$ experts indexed by $i = 1, \ldots, N$ and a set of tokens processed by routers across all layers in one mini-batch $B$ denoted as $\mathbf{Q}$, the load balancing loss for a transformer block containing an MoE layer is defined as:

$$\mathcal{L}_{\text{balance}} = N \cdot \sum_{i=1}^{N} f_i \cdot P_i, \quad f_i = \frac{1}{|\mathbf{Q}|} \sum_{x \in \mathbf{Q}} \mathbb{1}\{ i \in \text{TopK}(p(x)) \}, \quad P_i = \frac{1}{|\mathbf{Q}|} \sum_{x \in \mathbf{Q}} p_i(x), \qquad (3)$$

where $f_i$ is the fraction of tokens assigned to expert $i$, $p(x) = \text{Softmax}(x \cdot W)$ denotes the router probability of token $x$ across $N$ experts (without applying TopK), and $P_i$ is the fraction of the router probability allocated to expert $i$.

## 4 METHODOLOGY

This section describes the proposed CLIP-FMoE method. We outline the complete training pipeline and then provide a stage-wise explanation of our approach, highlighting the use of Isolated Constrained Contrastive Learning and Fusion Gate model design.

### 4.1 TRAINING PIPELINE

Recently, Sparse Upcycling (Komatsuzaki et al., 2023) was proposed to initialize MoE models by replicating the MLP layers of a pretrained dense model into multiple experts, thereby enabling efficient model scaling without training from scratch. However, the replication of MLPs hinders the development of expert specialization, which in turn restricts the model's ultimate capacity and produces suboptimal performance (He et al., 2024; Jiang et al., 2025; Nakamura et al., 2025).

To address this limitation, CLIP-MoE (Zhang et al., 2024c) adopts the Multi-stage Contrastive Learning (MCL) (Zhang et al., 2024b) strategy, which trains MLP series sequentially before integrating them into a unified MoE model. Specifically, this approach encourages each expert to learn distinct feature representations at each stage through an inheritance mechanism, where embedding clusters from the previous step determine the negative samples for the subsequent stage.

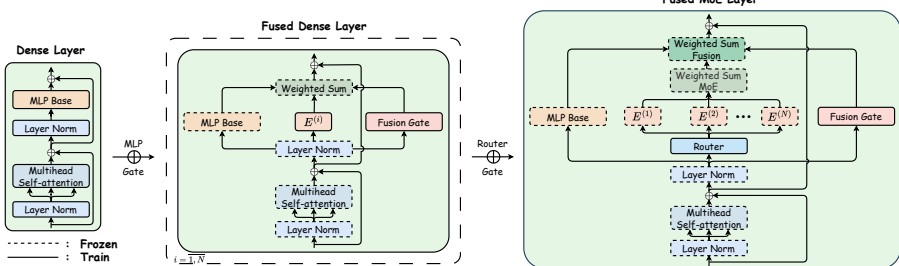

Figure 2: Mixture-of-experts architecture pipeline. Vanilla Transformer Block (left) is initialized from the pretrained model. Each Expert and Fusion Gate in $N$ models (middle) is fine-tuned. $N$ experts are aggregated in a unified block (right) while the Router and Fusion Gate are fine-tuned. Layers with solid borders are trainable, while layers with dashed borders remain frozen.

Samples belonging to the same cluster in the previous step are assumed to represent the same feature type and are assigned as negative samples in the next stage to facilitate learning of new feature representations. However, this design suffers from accumulated clustering errors and computational inefficiency in sequential learning. To enable efficient MoE scaling while ensuring expert specialization, we introduce CLIP-FMoE, a two-stage training paradigm, illustrated in Figure 1.

**Stage 1: Isolated Constrained Contrastive Learning.** We apply an ICCL mechanism by performing two-level semantic clustering to create specialized training data subsets for each MLP series. An MLP series comprises a set of MLPs from multiple layers of the base model that are jointly fine-tuned on the same data subset. These MLP series are then trained independently and in parallel to ensure expert diversification.

**Stage 2: Unification of Domain-specific Experts.** We integrate all trained MLP series into a unified MoE model as domain-specific experts through lightweight training of the router and fusion gate components. This approach enables efficient consolidation of specialized knowledge while maintaining computational efficiency.

## 4.2 ISOLATED CONSTRAINED CONTRASTIVE LEARNING

**Data semantic clustering.** At the beginning of Stage 1, we use the original base model to extract feature embeddings from the entire dataset for semantic clustering. This choice of embedding model is motivated by the intuition that the base model's representations can help identify challenging domains where its features tend to collapse or become indistinguishable. In the context of CLIP, we can utilize both text and image modalities for embeddings. However, based on experimental observations, we use the CLIP image encoder for all settings due to its superior robustness.

In the first level of clustering, we apply $K$-means to the embedding features of the original dataset $\mathcal{D}$, partitioning it into $N$ semantic clusters $\{C_i\}_{i=1}^N$ that potentially capture coarse-grained attributes useful for expert specialization. At the second level, we further apply $K$-means within each cluster $C_i$, dividing it into $M$ sub-clusters $\{S_j^{(i)}\}_{j=1}^M$ to capture more fine-grained semantic distinctions.

**Model initialization.** For each cluster $C_i$, we fine-tune $|\mathcal{A}|$ dense layers of the image encoder and $|\mathcal{B}|$ dense layers of the text encoder, where $\mathcal{A}$ and $\mathcal{B}$ are sets of chosen layer indices. We denote the resulting MLP networks as $\{E_k^{(i)}\}_{k \in \mathcal{A} \times \mathcal{B}}$. In Stage 1, to encourage each MLP network $E_k^{(i)}$ to learn complementary knowledge that builds upon pre-trained representations rather than redundant features, we leverage an additional component called the *Fusion Gate*, which is introduced in the model architecture section. As illustrated in Figure 2, for each $k$-th layer in both the image and text encoders, the fused dense layer is initialized from the corresponding dense layer in the base model by adding a replicated base MLP ($E_k^{(i)}$) and a newly initialized *Fusion Gate*. During training, we keep only the MLP series $\{E_k^{(i)}\}_{k \in \mathcal{A} \times \mathcal{B}}$ and their corresponding gates trainable, while all other parameters remain frozen.

**Contrastive learning.** Following semantic clustering and model initialization, we train $N$ MLP networks in parallel, independently of each other, each in one of the $N$ clusters. By performing

| Model | Cars | Coun | FGVC | Food | Sun | Cal | Euro | Flow | Pets | IN-1K | IN-v2 | Avg. |
|---|---|---|---|---|---|---|---|---|---|---|---|---|
| OpenAI CLIP | 77.93 | 31.88 | 31.77 | 93.07 | 67.56 | 83.27 | 62.59 | 79.20 | 93.21 | 75.54 | 69.84 | 69.62 |
| Fine-tuning | 69.12 | 25.43 | 25.83 | 91.34 | 71.17 | 79.67 | 65.20 | 69.56 | 90.24 | 72.89 | 66.39 | 66.08 |
| Up-cycling | 68.98 | 25.36 | 27.66 | 90.78 | 70.82 | 81.22 | **69.98** | 68.52 | 91.11 | 72.92 | 66.27 | 66.69 |
| CLIP-MoE | 71.20 | 25.31 | 27.81 | 91.27 | 71.01 | 79.19 | 65.13 | 69.90 | 90.35 | 73.19 | 66.86 | 66.47 |
| CLIP-FMoE | **74.62** | **28.02** | **29.58** | **92.32** | **71.86** | **81.79** | 68.31 | **73.54** | **91.58** | 74.87 | **68.66** | **68.65** |

Table 1: Zero-shot classification results across common datasets. All methods use CLIP ViT-L/14, trained on CC3M (LLaVA-ReCap) dataset. The reported metric is top-1 accuracy. **Bold** denotes the best performance among CLIP-finetuning methods.

isolated training in each cluster $C_i$, each MLP series $\{E^{(i)}\}$ is guaranteed to learn diverse, non-overlapping knowledge representations, thereby achieving specialization within its domain.

To leverage the fine-grained semantic clustering within sub-clusters $\{S_j^{(i)}\}_{j=1}^M$ of $C_i$, we constrain the batch sampling process during contrastive learning to ensure all samples within a batch belong to the same sub-cluster, denoted as *Constrained Sampler* (see Appendix C for more details) shown in Figure 1. This approach contrasts with vanilla training processes that employ random sampling, which can compromise the quality of in-batch negatives. Zheng et al. (2024); Xiao et al. (2025) demonstrated the benefits of matching images with multiple corresponding captions. We adopt this strategy by computing contrastive loss between images and their associated caption sets, which contain various captions (*e.g.*, split from longer descriptions).

Given varying caption quality, we use weighted combinations across loss terms. For each image paired with $L$ positive captions, let $\{\gamma_c\}_{c=1}^L$ be the corresponding weights, where $\gamma_c \geq 0$ and $\sum_{c=1}^L \gamma_c = 1$. We then aggregate the contrastive losses across all positive captions bidirectionally, computing both image-to-caption and caption-to-image similarities to obtain final objective for Stage-1 training as follows:

$$\mathcal{L}_{\text{multi-pos}} = \tfrac{1}{2}\Big(\sum_{c=1}^L \gamma_c \, \text{InfoNCE}\big(I, T^c\big) + \sum_{c=1}^L \gamma_c \, \text{InfoNCE}\big(T^c, I\big)\Big), \tag{4}$$

where an embedding batch $\{I_i, T_i\}_{i=1}^B$ is sampled from the same sub-cluster.

### 4.3 UNIFICATION OF DOMAIN-SPECIFIC EXPERTS

**Knowledge-Preserving Fusion Mechanism.** We introduce the *Fusion Gate* mechanism that integrates pre-trained knowledge with the newly learned, domain-specific information from each semantic cluster $C_i$. The mechanism is applied in both training stages, as illustrated in Figure 2. In Stage 1, the fusion occurs within the representation space of the MLP series and their base counterparts $E_{base}$, which function as the experts denoted $E_{new}$ in Stage 2. In Stage 2, the fusion is between the base MLP, which represents pre-trained knowledge, and a mixture of domain-specific experts from the MLP series (also denoted as $E_{new}$, known as $E_{moe}$). Through this two-stage process, the *Fusion Gate* enables experts to not only inherit and effectively align with pre-trained knowledge in the first stage, but also be further enhanced by the MoE architecture in the second stage. The fusion is controlled by a learnable gating network producing $G(x) \in [0,1]^D$:

$$E_{agg}(x) = G(x)E_{base}(x) + (1 - G(x))E_{new}(x), \quad G(x) = \sigma(\textit{Linear}(x)),$$

where $\sigma$ is sigmoid and *Linear* maps tokens to $D$-dimensional gating coefficients. The layer output $E_{agg}(x)$ combines pretrained and domain-specific knowledge, then is added to the attention block's residual connection.

**Fused MoE.** After Stage 1, we obtain set of MLP series $\{ E_k^{(i)} \mid k \in \mathcal{A} \times \mathcal{B}, \ i \in [N] \}$. The process of Fused MoE initialization, MoE along with Fusion Gate, is described as follows: from a base CLIP model, specifically, for a chosen Transformer layer $k$ in either image or text encoder, we initialize $N$ experts $E_k^{(1)}, \ldots, E_k^{(N)}$ by copying the weights from Stage 1, and initialize the MoE router from scratch. While Fusion Gate is initialized from average weights of $\left\{ G_k^{(i)} \right\}_{i=1}^N$ in Stage 1 to connect

| Model | MS COCO | | | | Flickr 8K | | | | Flickr 30K | | | | Avg. |
|---|---|---|---|---|---|---|---|---|---|---|---|---|---|
| | IR | | TR | | IR | | TR | | IR | | TR | | |
| | R@1 | R@5 | R@1 | R@5 | R@1 | R@5 | R@1 | R@5 | R@1 | R@5 | R@1 | R@5 | |
| OpenAI CLIP | 36.51 | 61.01 | 56.38 | 79.36 | 61.60 | 86.36 | 77.40 | 94.00 | 65.22 | 87.26 | 85.10 | 97.30 | 73.96 |
| Fine-tuning | 49.04 | 73.36 | 63.76 | 83.94 | 73.24 | 93.14 | 83.20 | 96.80 | 78.16 | 94.85 | 90.90 | 99.00 | 81.62 |
| Up-cycling | 48.83 | 73.19 | 63.12 | 84.36 | 73.10 | **93.48** | 83.50 | 97.00 | 77.98 | 94.66 | 91.20 | 98.90 | 81.61 |
| CLIP-MoE | 49.46 | **73.80** | 65.30 | 85.44 | 73.28 | **93.48** | 85.70 | 96.90 | 78.22 | **94.94** | 92.00 | 99.00 | 82.29 |
| CLIP-FMoE | **49.60** | 73.69 | **66.66** | **86.84** | **73.68** | 93.40 | **86.10** | **97.40** | **78.42** | 94.58 | **92.30** | **99.50** | **82.68** |

Table 2: Standard image-text retrieval results (R@1, R@5) on MS COCO, Flickr8K, and Flickr30K benchmarks. All methods use CLIP ViT-L/14 architecture with 77-token context length, trained on CC3M with LLaVA-generated captions. **Bold** denotes the best performance.

base MLP with a mixture of domain-specific experts, as illustrated in Figure 2. In this stage, only Fusion Gate and the MoE router are set to learnable, which not only keeps the specialization of experts but also is an efficient training strategy. In Stage 2, the Fused MoE model is trained on the entire dataset without constraints on sample size or batch sampling strategy. As in Stage 1, we train the model using the Multi-Positive InfoNCE loss, together with the token balance regularization defined in equation 3. The final objective is given by

$$\mathcal{L}_{\text{moe}} = \mathcal{L}_{\text{multi-pos}} + \alpha \mathcal{L}_{\text{balance}}, \tag{5}$$

where we set $\alpha = 0.01$ for all experiments.

## 5 EXPERIMENTS

### 5.1 MAIN RESULTS

#### 5.1.1 EXPERIMENTAL SETTINGS

**Dataset.** For our main experiments, we fine-tune a pretrained OpenAI CLIP ViT-L/14 model on LLaVA-ReCap variant of the CC3M dataset (Sharma et al., 2018; Li et al., 2024). Due to the original CLIP model's context length limitation of 77 tokens, we split long captions that exceed this limit into multiple semantically coherent sub-captions (typically four per image, constructed at the sentence level). In addition, we incorporate the raw caption version of CC3M. These combined sources (a total of $L = 5$ captions) constitute the set of positive captions used in our training dataset.

**Implementation.** Following Wang et al. (2025b), we fine-tune the MLP layers at odd-numbered positions and in the second half of both encoders of CLIP ViT-L/14. The CLIP-FMoE model employs 4 experts, with 2 experts activated during both training and inference. In Stage 1, the training data are clustered into $N = 4$ primary clusters, each further subdivided into $M = 64$ sub-clusters, yielding a total of 256 sub-clusters. We use a batch size of 1024 and run all experiments on 8 NVIDIA A100 GPUs. Additional details are provided in the appendix.

**Baselines.** We choose Vanilla Fine-tuning, Up-cycling MoE (Komatsuzaki et al., 2023), and CLIP-MoE (Zhang et al., 2024c) as our baselines. All are fine-tuned from the pretrained CLIP ViT-L/14 using the same training objectives described in Section 4.2. For a fair comparison, we fine-tune or apply up-cycling only on the same MLP layers used in our proposed model. To match the number of our activated parameters during training and inference, Up-cycling MoE and CLIP-MoE are configured with 5 experts and a top-3 routing.

#### 5.1.2 ZERO-SHOT CLASSIFICATION

Table 1 presents the zero-shot accuracy of our CLIP-FMoE compared with OpenAI CLIP, vanilla fine-tuning, Sparse Up-cycling, and CLIP-MoE. The evaluation is conducted on 11 benchmarks, encompassing fine-grained classification (Stanford Cars (Krause et al., 2013), FGVC-Aircraft (Maji et al., 2013)), scene and remote sensing (SUN397 (Xiao et al., 2010), EuroSAT (Helber et al., 2019), Country-211 (Radford et al., 2021)), domain-specific tasks (Flowers102 (Nilsback & Zisserman,

| Model | MS COCO | | Flickr 8K | | Flickr 30K | | DOCCI-Long | | IIW-Long | | Urban | | DCI | | Avg. |
|---|---|---|---|---|---|---|---|---|---|---|---|---|---|---|---|
| | IR | TR | IR | TR | IR | TR | IR | TR | IR | TR | IR | TR | IR | TR | |
| OpenAI CLIP | 36.5 | 56.4 | 61.6 | 77.4 | 65.2 | 85.1 | 63.3 | 66.0 | 90.5 | 90.2 | 55.8 | 68.3 | 33.9 | 36.0 | 63.3 |
| Fine-tuning | 46.6 | 64.3 | 72.1 | 84.6 | 74.8 | 90.8 | 77.7 | 77.6 | 97.5 | 96.8 | 88.9 | 89.8 | 54.4 | 57.6 | 76.7 |
| LongCLIP | 47.0 | 63.3 | 72.5 | 83.4 | 76.1 | 89.3 | 78.8 | 66.7 | 97.2 | 94.5 | 86.1 | 82.4 | 52.7 | 45.3 | 73.9 |
| TULIP | 45.7 | 61.2 | 70.6 | 81.5 | 74.6 | 88.0 | 79.6 | 78.0 | 97.2 | 96.2 | 91.2 | 90.2 | 54.7 | 54.3 | 75.9 |
| CLIP-FMoE | 47.7 | 65.6 | 72.4 | 84.9 | 75.9 | 91.6 | 79.2 | 78.4 | 97.3 | 98.0 | 88.2 | 90.7 | 54.7 | 55.9 | 77.2 |

Table 3: Retrieval performance on long-context scenarios. All methods use CLIP ViT-L/14 with 248-token context (except OpenAI CLIP), trained on ShareGPT4V. **Bold** indicates best performance, underline indicates second best.

2008), Oxford-IIIT Pets (Parkhi et al., 2012)), and general-purpose datasets (Caltech101 (Li et al., 2022), ImageNet-1K (Russakovsky et al., 2015), and ImageNet-v2 (Recht et al., 2019)).

Vanilla fine-tuning suffers from significant catastrophic forgetting, with its average accuracy dropping by nearly 4% compared to the original CLIP. This is especially evident on fine-grained datasets, such as a 25.83% accuracy on FGVC-Aircraft. In contrast, both Sparse Up-cycling and CLIP-MoE mitigate this issue, achieving average accuracies around 66.5%. However, both methods still underperform the original CLIP model by an average of approximately 3%.

On the other hand, CLIP-FMoE nearly closes the performance gap, reducing the average accuracy drop to less than 1% compared to the original CLIP. Our method consistently achieves the highest accuracy among all adaptation methods across every benchmark. These results demonstrate that the fusion-gate mechanism in CLIP-FMoE is highly effective at preserving pretrained knowledge.

### 5.1.3 ZERO-SHOT RETRIEVAL

We further evaluate our method on retrieval tasks using several established benchmarks: the standard image–text datasets MS-COCO (Lin et al., 2014) and Flickr30K (Plummer et al., 2015); the fine-grained retrieval sets DOCCI (Onoe et al., 2024) and IIW (Bell et al., 2014); and the long-caption datasets Urban-1K (Zhang et al., 2024a) and DCI (Urbanek et al., 2024), following the protocol of (Xiao et al., 2025). Table 2 and Table 8 in Appendix A together demonstrate that CLIP-FMoE consistently achieves the best performance across diverse settings and attains the highest average scores on both standard and fine-grained retrieval. Therefore, we hypothesize that the fusion gate goes beyond simple combination: for each token, it dynamically balances pretrained representations and domain-specific expert adaptations, utilizing whichever is more effective. Furthermore, this process is enhanced by our ICCL mechanism, which sharpens expert specialization on semantically coherent clusters. The combination of these components allows CLIP-FMoE to strike an effective balance, maintaining the pre-trained knowledge generalization while boosting performance on retrieval tasks.

### 5.2 LONG CONTEXT UNDERSTANDING

In this experiment, we evaluate the long context understanding capabilities of our proposed method. To handle longer captions, we follow LongCLIP (Zhang et al., 2024a) and extend the original context length of the base ViT-L/14 text encoder from 77 to 248 tokens by stretching positional embeddings through linear interpolation. For a fair comparison with the existing LongCLIP (Zhang et al., 2024a) and TULIP (Najdenkoska et al., 2025), we use ShareGPT4V (1.2M samples) (Chen et al., 2024) as our training dataset. Our experimental setup employs $N = 4$ clusters (equivalent to 4 experts) with 2 activated during both training and inference, $M = 16$ sub-clusters per cluster, and a batch size of 2048. We also fine-tune CLIP with extended context length as an additional baseline. Following LongCLIP's approach, we compute the objective loss using two versions of each caption: the full long caption and a shortened version consisting of only its first sentence. This results in a total of $L = 2$ captions per sample. We set $\gamma_1 = 0.1$ and $\gamma_2 = 0.9$. We evaluate on 7 datasets, comprising 3 standard image-text retrieval benchmarks (MS COCO, Flickr 8K, Flickr 30K) and 4 specialized long-caption retrieval datasets (DOCCI-Long, IIW-Long, Urban-1k, DCI).

Table 3 compares the retrieval performance on CLIP-FMoE with recent long-context methods. CLIP-FMoE outperforms others, with the highest average performance at 77.2%, outperforming the

| | Dataset | Ex. 1 | Ex. 2 | Ex. 3 | Ex. 4 | MoE |
|---|---|---|---|---|---|---|
| **Classification** | Cars | **62.95** | 61.56 | 58.72 | 58.08 | 61.15 |
| | Food-101 | 85.62 | **87.24** | 86.97 | 84.68 | **87.24** |
| | Sun397 | 66.73 | 66.11 | 65.63 | 66.12 | **67.87** |
| | Eurosat | 43.33 | 50.36 | 53.34 | 55.25 | **55.47** |
| | Caltech-101 | 82.38 | 80.92 | 82.81 | 82.79 | **82.95** |
| | Imagenet-a | **40.64** | 38.83 | 37.99 | 36.32 | 40.11 |
| | Imagenet-o | 41.65 | **43.35** | 42.4 | 41.55 | 42.55 |
| | Imagenet-r | **76.54** | 75.59 | 74.23 | 75.92 | 75.96 |
| | Imagenet-s | 46.58 | **46.84** | 46.04 | 46.43 | 46.61 |
| | Imagenet-1k | 64.45 | 64.75 | 65.36 | 64.85 | **65.99** |
| **Retrieval** | COCO IR | 39.48 | 39.92 | 39.55 | 41.82 | **42.18** |
| | COCO TR | 56.68 | 57.34 | 58.4 | **59.42** | 58.8 |
| | Urban IR | 80.6 | 84.4 | 84.4 | 86.2 | **87.2** |
| | Urban TR | 83.70 | 85.00 | 87.80 | 86.00 | **88.24** |
| | DCI IR | 47.69 | 49.25 | 48.81 | 51.84 | **51.98** |
| | DCI TR | 47.73 | 49.16 | 54.45 | 53.4 | **54.6** |

Table 4: Result of Experts and Unified Model across classification and retrieval datasets.

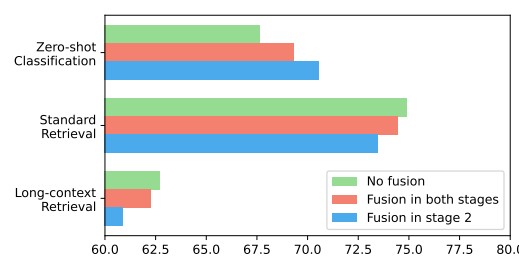

Figure 3: Ablation on Fusion Gate configurations and data variants. The $x$-axis represents the averaged Accuracy/R@1.

strong baseline TULIP by 1.3%. This consistent improvement across diverse datasets demonstrates the robustness of CLIP-FMoE, showing its effectiveness even in long-context settings.

## 5.3 EXPERT SPECIALIZATION

We present an analysis of expert-level performance within CLIP-FMoE alongside the unified model results. The evaluated model was trained on ShareGPT4V with an extended 248-token context length, built upon CLIP ViT-B/16. Individual expert evaluation was conducted by disabling the MoE routing mechanism and forcing all input tokens through a single expert. The evaluation protocol includes 10 zero-shot classification and 3 zero-shot retrieval datasets.

Table 4 demonstrates that CLIP-FMoE achieves meaningful expert specialization across domains. Expert 1 excels in automotive and adversarial scenarios (Cars, ImageNet-A/R), Expert 2 specializes in food recognition and out-of-distribution data (Food-101, ImageNet OOD variants), Expert 3 handles general classification (Caltech-101, ImageNet-1K), and Expert 4 dominates retrieval tasks (COCO IR/TR, Urban IR). The unified MoE model consistently matches or exceeds individual expert performance through dynamic expert combination. Crucially, experts learn complementary rather than redundant representations, with performance compensation between experts. This pattern confirms that ICCL effectively enhances model capability through expert diversification.

## 5.4 FUSION GATE ABLATION STUDY

To examine the role of Fusion Gate in CLIP-FMoE, we evaluate three variants: (1) no Fusion Gate, (2) Fusion Gate in both stages (standard design), and (3) Fusion Gate only in Stage 2. We report average performance across zero-shot classification (top-1 accuracy, 11 datasets from Table 1), standard retrieval (top-1 recall, 3 datasets from Table 2), and long-context retrieval (top-1 recall, 4 datasets from Table 8 in Appendix A). The results are shown in Figure 3.

Design (1) performs well on retrieval tasks but shows the weakest zero-shot classification, validating the necessity of Fusion Gate in preserving pretrained knowledge. Design (3) achieves superior classification but fails on diverse retrieval tasks, suggesting that Stage 2-only Fusion Gates prevent complementary expert learning and allow pre-trained representations to dominate. Our standard design (2) provides optimal balance across all tasks, effectively integrating with ICCL while maintaining expert specialization benefits.

## 5.5 EFFICIENCY AND SCALABILITY ANALYSIS

To evaluate the computational efficiency of our proposed CLIP-FMoE against the alternatives in Table 1, we report model size, per-sample FLOPs, trainable parameters per stage, clustering time (for CLIP-MoE and CLIP-FMoE), and total training time. All experiments were conducted under identical hardware, batch size, and precision settings. The results are summarized in Table 5.

| Model | # Parameters (M) | Cluster (min) | Total Training (min) | Inference (GFLOPs) |
|---|---|---|---|---|
| CLIP (Base) | 427.62 / 64.53 | - | 24.59 | 84.38 |
| Up-Cycling | 685.78 / 322.69 | - | 40.41 | 112.44 |
| CLIP-MoE | 685.78 / 64.53 / 0.04 | 60.32 | 193.51 | 112.44 |
| CLIP-FMoE | 693.83 / 72.59 / 8.09 | 14.54 | 53.24 | 128.22 |

Table 5: Efficiency comparison of CLIP (Fine-tuning), Up-Cycling, CLIP-MoE, and our CLIP-FMoE. Parameters are reported as either (base / trainable) or (base / trainable in Stage 1 / trainable in Stage 2), in millions (M). FLOPs are measured per sample (forward pass only) in GFLOPs. Training times are wall-clock measurements under identical hardware and training settings (total training time including clustering time).

CLIP-FMoE attains a favourable efficiency–performance trade-off: it reduces total training time by 72.5% compared with CLIP-MoE while increasing inference FLOPs by only 14%. Relative to Up-Cycling, CLIP-FMoE requires roughly 80% fewer peak trainable parameters. The resulting reduction in memory pressure and optimizer-state size improves optimization stability, making the modest inference overhead an acceptable compromise in scenarios where training time and GPU memory are the primary constraints.

## 6 CONCLUSION

This paper presents CLIP-FMoE, a novel CLIP scaling method that integrates Mixture-of-Experts architecture into base CLIP models. Our approach distinguishes itself from existing methods by preserving zero-shot capabilities through a Fusion Gate mechanism while ensuring expert specialization via an Isolated Constrained Contrastive Learning training pipeline. Experimental results demonstrate the effectiveness of CLIP-FMoE across diverse settings, including maintaining zero-shot classification performance and improving fine-grained and long-context understanding capabilities. However, we have limited understanding of how the semantic data clustering impacts final model performance. Future work could extend CLIP-FMoE to other multimodal alignment scenarios such as video-text, audio-text, or multilingual vision-language tasks.

## REPRODUCIBILITY STATEMENT

All the models and datasets used in experimental results are publicly available. We have described the detailed experimental settings includes the dataset, model settings, and baselines at the beginning of each section. Moreover, we provide detailed hyperparameter settings in Table 9 and 10 in the Appendix. The full code will be released upon acceptance.

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

## A  MAIN EXPERIMENTAL RESULTS

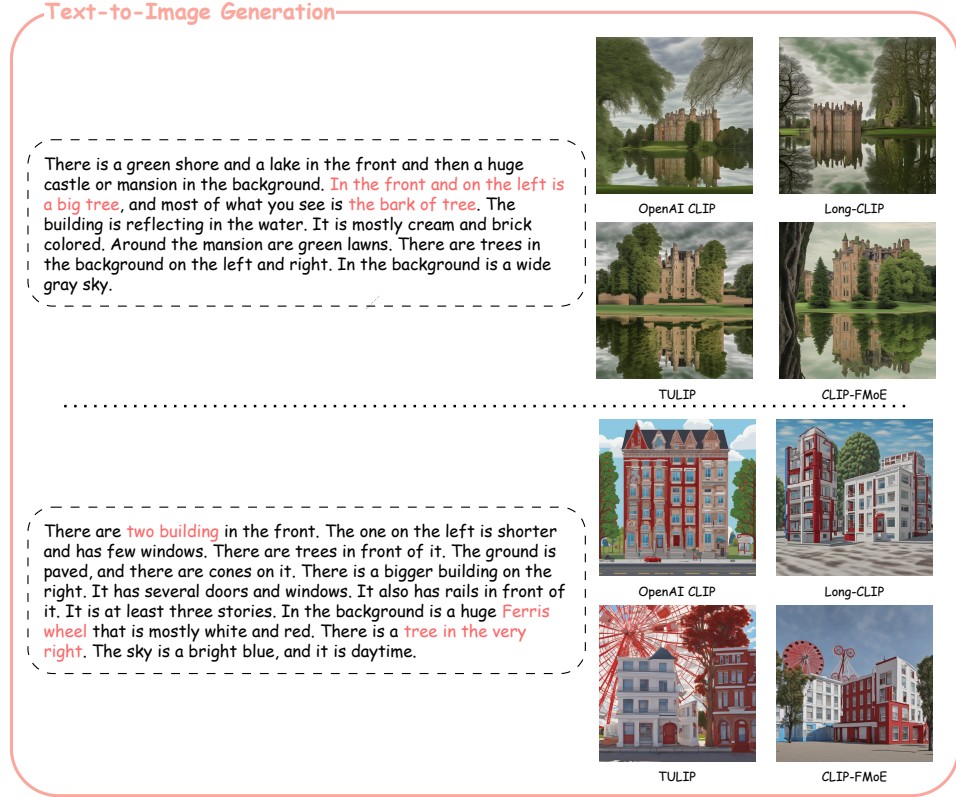

Figure 4: Generated image samples from Stable-Diffusion-XL, conditioned on the output of the text encoder. Highlighted texts indicate the aspects that our method successfully captures compared to the baselines.

| Model | Cars | Coun | FGVC | Food | Sun | Cal | C10 | C100 | Euro | Flow | Pets | IN-a | IN-o | IN-r | IN-1K | IN-Ske | IN-v2 | Avg. |
|---|---|---|---|---|---|---|---|---|---|---|---|---|---|---|---|---|---|---|
| CLIP OpenAI | 77.93 | 31.88 | 31.77 | 93.07 | 67.56 | 83.27 | 95.59 | 75.85 | 62.59 | 79.20 | 93.21 | 70.75 | 32.25 | 87.82 | 75.54 | 59.58 | 69.84 | 69.86 |
| Fine-tuning | 69.12 | 25.43 | 25.83 | 91.34 | 71.17 | 79.67 | **96.63** | 77.93 | 65.20 | 69.56 | 90.24 | 61.92 | 35.30 | 85.62 | 72.89 | 56.37 | 66.39 | 67.09 |
| Up-cycling | 68.98 | 25.36 | 27.66 | 90.78 | 70.82 | 81.22 | 95.72 | 77.92 | **69.98** | 68.52 | 91.11 | 61.43 | 35.55 | 85.73 | 72.92 | 56.62 | 66.27 | 67.45 |
| CLIP-MoE | 71.20 | 25.31 | 27.81 | 91.27 | 71.01 | 79.19 | 95.82 | 77.70 | 65.13 | 69.90 | 90.35 | 62.33 | 34.75 | 86.23 | 73.19 | 56.89 | 66.86 | 67.35 |
| CLIP-FMoE | **74.62** | **28.02** | **29.58** | **92.32** | **71.86** | **81.79** | 95.74 | **78.75** | 68.31 | **73.54** | **91.58** | **66.13** | **35.80** | **87.46** | **74.87** | **58.71** | **68.66** | **69.28** |

Table 6: Top-1 accuracy across 17 benchmark datasets (extended version of Table 1). Models trained on LLaVA-Recap CC3M with ViT-L/14 backbone (77-token context). **Bold** denotes the best method out of reported fine-tuning methods

### A.1  DATASET

For the main results, we use the LLaVA-Recap CC3M dataset (Li et al., 2024), which contains approximately 2.8M samples with long captions. For each image, the set of positive captions consists of one raw caption from the original CC3M dataset and four detailed captions.

The detailed captions are generated as follows: we first split each long caption from the LLaVA-Recap CC3M dataset into sentences and compute the token count for each sentence. We then randomly combine sentences to construct detailed captions of approximately 77 tokens each. Although we initially generate five detailed captions, we retain only four to avoid potential sentence duplication in the final set. The complete process will be documented in the code repository to be released later.

The weights of the loss term $\gamma$ are set to 0.1 for the raw caption and 0.225 for each of the four detailed captions, encouraging greater diversity.

| Model | Cars | Coun | FGVC | Food | Sun | Cal | C10 | C100 | Euro | Flow | Pets | IN-a | IN-o | IN-r | IN-1K | IN-Ske | IN-v2 | **Avg.** |
|---|---|---|---|---|---|---|---|---|---|---|---|---|---|---|---|---|---|---|
| OpenAI CLIP | 77.93 | 31.88 | 31.77 | 93.07 | 67.56 | 83.27 | 95.59 | 75.85 | 62.59 | 79.20 | 93.21 | 70.75 | 32.25 | 87.82 | 75.54 | 59.58 | 69.84 | 69.86 |
| Fine-tune | 74.36 | 28.68 | 28.18 | 91.00 | 70.27 | **85.00** | 94.74 | 76.18 | 61.28 | 72.82 | 91.17 | 59.41 | 35.60 | 85.56 | 71.69 | 57.74 | 65.53 | 67.60 |
| LongCLIP | 72.37 | 28.14 | **30.78** | 91.47 | **72.54** | 82.58 | **96.10** | **79.82** | **69.09** | 72.39 | 91.03 | 64.13 | **37.00** | 86.44 | 72.82 | 57.95 | 66.88 | 68.91 |
| TULIP | 63.25 | 21.75 | 19.95 | 83.46 | 67.24 | 83.06 | 92.96 | 71.91 | 55.43 | 61.64 | 87.03 | 55.28 | 33.90 | 81.42 | 63.48 | 52.51 | 57.56 | 61.87 |
| CLIP-FMoE | **75.58** | **30.17** | 29.07 | **92.32** | 71.54 | 84.63 | 95.26 | 78.02 | 67.11 | **76.06** | 93.08 | **65.24** | 35.90 | **87.27** | **73.90** | **58.84** | **67.72** | **69.51** |

Table 7: Top-1 accuracy across 17 common benchmark datasets. Results are obtained using models trained on ShareGPT4V dataset with ViT-L/14 architecture and a standard 248-token context length.

| Model | DOCCI | | IIW | | Urban | | DCI | | Avg. |
|---|---|---|---|---|---|---|---|---|---|
| | IR | TR | IR | TR | IR | TR | IR | TR | |
| OpenAI CLIP | 17.1 | 48.5 | 35.0 | 84.3 | 55.8 | 68.3 | 33.9 | 36.0 | 47.4 |
| Vanilla Fine-tuning | 23.9 | 53.6 | 46.8 | 88.5 | 87.3 | 85.9 | 51.1 | 49.2 | 60.8 |
| CLIP Up-cycling | 23.9 | 52.9 | 46.6 | 89.0 | **87.6** | 85.4 | 53.1 | 51.3 | 61.2 |
| CLIP-MoE | **24.1** | 56.1 | **47.2** | 89.8 | 86.1 | **86.0** | 53.5 | 52.0 | 61.8 |
| CLIP-FMoE | 23.7 | **58.2** | 45.4 | **92.5** | 85.5 | 85.7 | **53.9** | **53.0** | **62.2** |

Table 8: R@1 retrieval results on fine-grained (DOCCI, IIW) and long-caption (Urban-1k and DCI) datasets.

## A.2 EXPERIMENTAL SETUP

The main experiment uses the OpenAI CLIP ViT-L/14 (Radford et al., 2021) model; we use standard context length 77 tokens. We apply CLIP-FMoE (Zhang et al., 2024c) only to odd layers in the last half of both encoders to reduce computational requirements and demonstrate the lightweight nature of our method. Detailed hyperparameters are provided in Table 9.

## A.3 RESULTS

We present extended zero-shot classification results across 17 datasets (compared to the condensed results in the main paper). These results demonstrate CLIP-FMoE's consistent knowledge-preserving ability across a wide range of benchmark datasets, outperforming existing methods by significant margins.

## B LONG CONTEXT UNDERSTANDING

| Hyperparameter | Value |
|---|---|
| Global batch size | 1024 |
| Epochs (per-expert) | 1 |
| Epochs (full MoE) | 1 |
| Positive captions ($L$) | 5 |
| Warm-up steps | 3% total steps |
| Learning rate | $5 \times 10^{-5}$ |
| Optimizer | AdamW |
| $\quad \beta_1$ | 0.9 |
| $\quad \beta_2$ | 0.999 |
| $\quad \epsilon$ | $1 \times 10^{-8}$ |
| $\quad$ Weight decay | 0.5 |
| Precision | AMP mixed |
| Data loader workers | 16 |
| Experts per cluster ($N$) | 4 |
| Sub-clusters per cluster ($M$) | 16 |

Table 9: Hyperparameters for CC3M.

| Hyperparameter | Value |
|---|---|
| Global batch size | 2048 |
| Epochs (per-expert) | 1 |
| Epochs (full MoE) | 1 |
| Positive captions ($L$) | 2 |
| Warm-up steps | 3% total steps |
| Learning rate | $5 \times 10^{-5}$ |
| Optimizer | AdamW |
| $\quad \beta_1$ | 0.9 |
| $\quad \beta_2$ | 0.999 |
| $\quad \epsilon$ | $1 \times 10^{-8}$ |
| $\quad$ Weight decay | 0.5 |
| Precision | AMP mixed |
| Data loader workers | 16 |
| Experts per cluster ($N$) | 4 |
| Sub-clusters per cluster ($M$) | 16 |

Table 10: Hyperparameters for Long Context.

### B.1 DATASET

We use the ShareGPT4V dataset (Chen et al., 2024) for this experiment. Following LongCLIP's experimental setting, the positive caption set for each image consists of the original long caption and its first sentence. We set the loss weights to 0.1 and 0.9 respectively for these two caption types.

### B.2 EXPERIMENTAL SETUP

The long context understanding experiment uses the OpenAI CLIP ViT-L/14 model. Following LongCLIP, we extend the context length from 77 to 248 tokens. We apply CLIP-FMoE only to odd layers in the last half of both encoders to reduce computational requirements and demonstrate the lightweight nature of our method. Detailed hyperparameters are provided in Table 10.

### B.3 RESULTS

Table 7 presents the zero-shot classification results for each method in the long context understanding experiment. These results indicate that other long-context understanding methods also experience decreased generalization performance when trained on new data, such as long-caption datasets, despite achieving improvements on benchmarks involving longer input captions. In contrast, CLIP-FMoE performs well in all benchmarks compared to TULIP (Najdenkoska et al., 2025) and Long-CLIP (Zhang et al., 2024a), which rely on more complex training techniques such as coarse-grained feature and short caption alignment learning or text encoder distillation, while our approach employs a straightforward training objective. This suggests that our method has potential for integration with other techniques to enhance CLIP's capabilities.

### B.4 STABLE DIFFUSION QUANTITATIVE RESULTS

Following the approach of LongCLIP (Zhang et al., 2024a), we integrated our trained CLIP Text Encoder into Stable Diffusion XL (Podell et al., 2024) as its text encoding component. The results are presented in Figure 4.

The figure demonstrates that CLIP-FMoE exhibits superior performance compared to previous long-context approaches in capturing fine-grained textual details during image generation.

## C DETAIL MECHANISM OF CONSTRAINED SAMPLER

In Stage 1, for each cluster $C_i$ containing $M$ sub-clusters $\{S_j^{(i)}\}_{j=1}^M$, we fine-tune only the MLP (FFN) modules at a selected set of layer indices: $\mathcal{A}$ in the image encoder and $\mathcal{B}$ in the text encoder.

At each training iteration the Constrained Sampler is responsible for producing a mini-batch size $B$ that contains only samples from a single sub-cluster $S_j \subseteq C_i$, as described in Zhang et al. (2024c). The sampling procedure is implemented as follows:

1. **Shuffle each sub-cluster.** We permute the indices of every sub-cluster $S_j$ using a fixed seed for reproducibility.

2. **Trim to full batches.** For each sub-cluster $S_j$ with $n_j = |S_j|$ samples, keep only the first $\lfloor n_j/B \rfloor \cdot B$ indices from the permuted order. This ensures each retained sub-cluster yields an integer number of full-size mini-batches and avoids partial batches.

3. **Batch Sampling.** Cycle through the trimmed sub-clusters in round-robin order. For each selected sub-cluster, pop samples and append them to the current mini-batch until the batch reaches size $B$. When a full batch is assembled, yield it and proceed to the next sub-cluster. Continue until no sub-cluster can produce another full batch.

## D ABLATION STUDY OF CLUSTERING TYPES

To better understand how clustering types affect final performance, we examine several clustering variants beyond our standard image-based clustering: (1) random clustering with only level 1, (2)

| Cluster Type | MS COCO | | | | Flickr 8K | | | | Flickr 30K | | | | Avg. |
|---|---|---|---|---|---|---|---|---|---|---|---|---|---|
| | Image Retrieval | | Text Retrieval | | Image Retrieval | | Text Retrieval | | Image Retrieval | | Text Retrieval | | |
| | R@1 | R@5 | R@1 | R@5 | R@1 | R@5 | R@1 | R@5 | R@1 | R@5 | R@1 | R@5 | |
| Random | 40.8 | 66.0 | 58.8 | 81.8 | 64.9 | 88.4 | 80.1 | 95.0 | 69.8 | 89.3 | 86.6 | 97.9 | 76.6 |
| Text | 41.8 | 66.9 | 59.8 | 82.4 | 65.6 | 88.6 | 80.5 | 95.3 | **70.8** | 90.3 | 87.5 | 98.2 | 77.3 |
| Image + Text | 41.1 | 66.5 | 59.6 | 82.3 | 65.4 | 88.3 | 80.6 | 95.3 | 70.7 | 89.9 | **89.2** | **98.6** | 77.3 |
| Image (level 1) | 41.3 | 66.6 | 59.8 | **82.8** | 65.9 | 88.4 | 80.7 | 95.3 | 70.6 | 90.3 | 87.5 | 98.5 | 77.3 |
| Standard | **42.2** | **67.3** | **60.3** | 82.5 | **66.0** | **89.0** | **80.9** | **95.4** | 70.7 | **90.5** | 88.6 | 98.5 | **77.7** |

Table 11: Standard text-image retrieval results for clustering method ablation study of CLIP-FMoE. All experiments were conducted with a 248-token context length and ViT-B/16 architecture on ShareGPT4V dataset.

| Cluster Type | Fine-grained Text-Image Retrieval | | | | | | | | Long Text-Image Retrieval | | | | | | | | Avg. |
|---|---|---|---|---|---|---|---|---|---|---|---|---|---|---|---|---|---|
| | DOCCI-FG | | | | IIW-FG | | | | Urban 1K | | | | DCI | | | | |
| | IR | | TR | | IR | | TR | | IR | | TR | | IR | | TR | | |
| | R@1 | R@5 | R@1 | R@5 | R@1 | R@5 | R@1 | R@5 | R@1 | R@5 | R@1 | R@5 | R@1 | R@5 | R@1 | R@5 | |
| Random | 15.7 | 29.8 | 47.1 | 72.9 | 32.2 | 52.9 | 88.0 | 98.0 | 84.5 | 96.7 | 87.0 | 97.0 | 50.4 | 69.2 | 52.2 | 71.0 | 65.3 |
| Text | 16.0 | **30.4** | 47.8 | **74.8** | **32.6** | 52.9 | 87.5 | **98.5** | 86.1 | **97.7** | **89.9** | 98.6 | 51.5 | **70.5** | 53.3 | 71.8 | 66.2 |
| Image + Text | 15.9 | 30.0 | **48.0** | 73.8 | 32.3 | 52.0 | **89.8** | 98.3 | 85.5 | 96.9 | 88.1 | 98.3 | 50.9 | 69.6 | 53.4 | 71.9 | 65.9 |
| Image (level 1) | 16.1 | 30.2 | 47.4 | 73.8 | 32.6 | 52.2 | 88.5 | 98.0 | 85.8 | 96.1 | 87.9 | 98.5 | 51.6 | 69.7 | 54.0 | 72.7 | 66.0 |
| Standard | **16.1** | 30.3 | 47.9 | 74.1 | 32.6 | 52.4 | 88.0 | 98.0 | **87.2** | 97.1 | 88.2 | **98.6** | **52.0** | 70.3 | **54.6** | **73.2** | 66.3 |

Table 12: Fine-grained and long-context retrieval results for clustering method ablation study of CLIP-FMoE. All experiments were conducted with a 248-token context length and ViT-B/16 architecture on ShareGPT4V dataset.

text-based clustering, (3) multimodal clustering using both image and text features, and (4) image clustering with only level 1. Note that standard, text-based, and multimodal clustering employ the two-level clustering approach described in the main paper. All embedding steps are performed using the same OpenAI CLIP ViT-B/16 model used for training. For multimodal clustering, we use a Cartesian product approach. For example, to obtain 4 clusters, we first cluster data based on text features into 2 clusters and separately cluster based on image features into 2 clusters, then use their Cartesian product to form 4 multimodal clusters. This is analogous to performing level 2 clustering on each cluster to obtain sub-clusters. For this ablation study, we use OpenAI CLIP ViT-B/16 as the base model, with all other configurations identical to those listed in Table 10.

Results from Tables 11 and 12 indicate that our standard clustering approach achieves the best performance. Random clustering, while being a naive divide-and-conquer approach, cannot ensure high-quality in-batch negatives. Similarly, image clustering with only level 1 may achieve better performance by directing experts to learn distinct domains as categorized by the base model, but it fails to ensure in-batch negative quality due to free sampling in the contrastive learning objective. For multimodal and text-based clustering, performance remains worse than image-only clustering. This may be attributed to the limited representation quality of CLIP's text encoder, which was originally trained on web-crawled text captions rather than high-quality corpora used in modern LLMs. In our current work, motivated by enabling experts to learn challenging domains identified by the base model, we utilize the base model itself for clustering. However, alternative embedding models could also be explored for this purpose.

## E  ANALYSIS ON FINE-TUNING DATASET

In this section, we provide analysis on the choice of fine-tuning dataset. We compare the classification and retrieval performance between fine-tuning on ShareGPT4V (Chen et al., 2024) and LLaVA-Recap (Li et al., 2024). The results are presented in Tables 13 and 14. The results show that fine-tuning on LLaVA-Recap dataset yield better performance in both tasks, showcasing the benefits of scaling the number of data samples, and detailed description.

## F  ANALYSIS ON THE NUMBER OF CLUSTERS (EXPERTS) AND SUB-CLUSTERS

We conduct experiments to investigate the impact of the number of clusters (experts) $N$ and the number of sub-clusters $M$. The results are presented in Tables 15 and 16. Our analysis reveals that

| Data type | MS COCO | | Flickr 8K | | Flickr 30K | | DOCCI-Long | | IIW-Long | | Urban | | DCI | | Avg. |
|---|---|---|---|---|---|---|---|---|---|---|---|---|---|---|---|
| | IR | TR | IR | TR | IR | TR | IR | TR | IR | TR | IR | TR | IR | TR | |
| 400M OpenAI | 36.51 | 56.38 | 61.60 | 77.40 | 65.22 | 85.10 | 63.25 | 66.04 | 90.50 | 90.25 | 55.80 | 68.30 | 33.94 | 36.04 | 63.31 |
| + 1.2M ShareGPT4V | 47.57 | 65.48 | 72.08 | 84.90 | 75.98 | 91.10 | 78.83 | 77.61 | 97.50 | **98.25** | 88.00 | 90.50 | 54.50 | 55.43 | 76.98 |
| + 2.8M LLaVA-Recap | **49.50** | **67.00** | **74.02** | **85.20** | **78.54** | **93.10** | **84.92** | **81.90** | **97.75** | 96.75 | **93.40** | **94.00** | **59.88** | **58.07** | **79.57** |

Table 13: Ablation study of CLIP-FMoE on standard text-image retrieval tasks with varying fine-tuning dataset sizes. Results are obtained using the ViT-L/14 architecture with a context length of 248.

| Data type | Cars | Coun | FGVC | Food | Sun | Cal | C10 | C100 | Euro | Flow | Pets | IN-a | IN-o | IN-r | IN-1K | IN-Ske | IN-v2 | Avg. |
|---|---|---|---|---|---|---|---|---|---|---|---|---|---|---|---|---|---|---|
| 400M OpenAI | 77.9 | 31.9 | 31.8 | 93.1 | 67.6 | 83.3 | 95.6 | 75.9 | 62.6 | 79.2 | 93.2 | 70.8 | 32.3 | 87.8 | 75.5 | 59.6 | 69.8 | 69.9 |
| + 1.2M ShareGPT4V | **75.8** | **30.1** | **29.8** | 92.4 | 65.1 | **84.4** | 95.4 | 78.3 | 67.2 | **76.0** | **93.0** | 65.1 | 35.6 | 87.2 | 73.9 | 58.9 | 67.8 | 69.2 |
| + 2.8M LLaVA-Recap | 74.6 | 28.0 | 29.7 | **92.8** | **71.9** | 82.3 | **95.7** | **79.0** | **68.0** | 74.2 | 92.2 | **66.1** | **35.8** | **87.4** | **74.7** | **59.0** | **68.7** | **69.4** |

Table 14: Ablation study of CLIP-FMoE Top-1 zero-shot classification accuracy across 17 benchmark tasks with varying fine-tuning dataset sizes. Results are reported using the ViT-L/14 architecture with a context length of 248.

| $M$ | $N$ | MS COCO | | Flickr 8K | | Flickr 30K | | Urban | | DCI | | Avg. |
|---|---|---|---|---|---|---|---|---|---|---|---|---|
| | | IR | TR | IR | TR | IR | TR | IR | TR | IR | TR | |
| 8 | 4 | 41.6 | 60.2 | 65.4 | 80.5 | 70.9 | 88.4 | 87 | 88 | 52.2 | 54.3 | 68.9 |
| | 8 | 40.6 | 59.4 | 64.8 | 79.7 | 70.1 | 86.9 | 83.5 | 86.3 | 50.8 | 53.2 | 67.5 |
| 16 | 4 | 42.2 | 60.3 | 66 | 80.9 | 70.7 | 88.6 | 87.2 | 88.2 | 52 | 54.6 | 69.1 |
| | 8 | 40.6 | 58.8 | 64.8 | 79.7 | 69.8 | 87.1 | 82.7 | 86.5 | 50.5 | 52.8 | 67.3 |
| 32 | 4 | 41.9 | 60.4 | 66 | 80.8 | 70.8 | 87.8 | 87.2 | 89.6 | 52.1 | 54.4 | 69.1 |
| | 8 | 40.5 | 59.2 | 64.7 | 79.6 | 69.6 | 87 | 83.3 | 87.5 | 50.7 | 52.8 | 67.5 |
| 64 | 4 | 42 | 60.9 | 66 | 81 | 70.8 | 88.1 | 87.7 | 90.2 | 52.9 | 55.1 | 69.5 |
| | 8 | 40.4 | 58.6 | 64.9 | 79.4 | 69.9 | 87.7 | 83.4 | 88.1 | 50.9 | 52.5 | 67.6 |

Table 15: Hyperparameter analysis on the number of clusters/experts ($N$) and the number of sub-clusters ($M$). The setting is CLIP-FMoE on standard text-image retrieval tasks. Results are obtained using the ViT-B/16 architecture with a context length of 248.

| $M$ | $N$ | Cars | Coun | FGVC | Food | Sun | Cal | C10 | C100 | Euro | Flow | Pets | IN-a | IN-o | IN-r | IN-1K | IN-Ske | IN-v2 | Avg. |
|---|---|---|---|---|---|---|---|---|---|---|---|---|---|---|---|---|---|---|---|
| 8 | 4 | 61.5 | 22 | 21.9 | 87.4 | 67.8 | 82.7 | 90.8 | 67.2 | 55.6 | 69.5 | 89.9 | 40.4 | 42.7 | 76.2 | 66.1 | 47.6 | 59.1 | 61.7 |
| | 8 | 61.5 | 22.2 | 22.5 | 88 | 67.6 | 82.7 | 90.8 | 68.4 | 57.9 | 68.8 | 89.3 | 41.8 | 44.3 | 76.4 | 66.5 | 47.6 | 59.7 | 62.1 |
| 16 | 4 | 61.2 | 22 | 22.2 | 87.9 | 67.9 | 82.8 | 90.6 | 66.6 | 55.7 | 69.5 | 89.6 | 40.2 | 42.6 | 76 | 66 | 47.5 | 58.9 | 61.6 |
| | 8 | 62.2 | 22 | 22.4 | 88.2 | 67.6 | 83 | 90.4 | 68.3 | 58.4 | 68.3 | 89.3 | 41.6 | 43.9 | 76.2 | 66.4 | 47.5 | 59.5 | 62.1 |
| 32 | 4 | 61.3 | 22 | 22.6 | 87.2 | 67.8 | 83.2 | 91.1 | 67.4 | 54.4 | 69.2 | 89.5 | 40.8 | 43.2 | 76.3 | 66.1 | 47.5 | 58.9 | 61.7 |
| | 8 | 61.9 | 21.9 | 22.3 | 88 | 67.4 | 82.8 | 90.8 | 67.7 | 59.4 | 67.8 | 88.9 | 41.32 | 44.2 | 76.4 | 66.2 | 47.4 | 59.3 | 62.0 |
| 64 | 4 | 61.5 | 21.5 | 21.8 | 87.1 | 67.6 | 82.9 | 91 | 67.4 | 56 | 68.9 | 89.2 | 40.6 | 42.6 | 76.2 | 65.7 | 47.2 | 58.5 | 61.5 |
| | 8 | 61.4 | 21.5 | 22.3 | 87.8 | 67.6 | 82.8 | 90.4 | 67.9 | 57.3 | 67.7 | 88.6 | 40.8 | 43.6 | 76.1 | 66.2 | 47.3 | 59.6 | 61.7 |

Table 16: Hyperparameter analysis on the number of clusters/experts ($N$) and the number of sub-clusters ($M$). The setting is CLIP-FMoE on zero-shot image classification task. Results are obtained using the ViT-B/16 architecture with a context length of 248.

increasing the number of experts $N$ leads to improved classification performance but diminished performance on retrieval tasks. We conclude that scaling the number of experts introduces a trade-off between classification and retrieval performance. The performance is robust with the choice of number of sub-clusters, with their results differs from each other in less than 1%.

## G    EXPERT SPECIALIZATION

To further investigate the expert specialization, we conduct an additional analysis on the MMVP benchmark (Tong et al., 2024) using the same experts (ViT-B/16) as in Table 4 and show that they capture diverse and complementary attributes. MMVP evaluates a CLIP model by asking it to select

| Attribute | Expert 0 | Expert 1 | Expert 2 | Expert 3 |
|---|---|---|---|---|
| Orientation and Direction | 13.33 | **20** | 0 | 6.67 |
| Presence of Specific Features | 6.67 | **20** | 13.33 | 6.67 |
| State and Condition | 40 | 40 | 40 | **46.67** |
| Quantity and Count | **20** | 13.33 | 13.33 | 13.33 |
| Positional and Relational Context | 0 | 0 | **20** | 6.67 |
| Color and Appearance | 53.33 | 66.67 | **73.33** | 66.67 |
| Structural Characteristics | 20 | 26.67 | **33.33** | 26.67 |
| Texts | 6.67 | 6.67 | **13.33** | **13.33** |
| Viewpoint and Perspective | 13.33 | 6.67 | **20** | 6.67 |

Table 17: Attribute distribution across experts for different characteristics.

the correct image, given a textual statement, from a pair of visually similar images. The evaluation set is carefully categorized by human annotators into nine fine-grained visual attributes.

The results in Table 17 indicate clear attribute-level specialization across experts. For example, Expert 0 performs best on Quantity and Count. Expert 1 excels in Orientation and Direction and Presence of Specific Features. Expert 2 focuses on Color and Appearance and Structural Characteristics, while Expert 3 is strongest on State and Condition and shares the lead on Texts. This pattern supports our claim that semantic clustering induces meaningful routing and complementary feature learning rather than redundant experts.

| Dataset | Cars | Food | Sun | Eurosat | Cal | IN-a | IN-o | IN-r | IN-s | IN-1K | IN-v2 | FLow | Avg. |
|---|---|---|---|---|---|---|---|---|---|---|---|---|---|
| Worst among experts | 65.69 | 88.49 | 64.71 | 48.83 | 75.22 | 54.88 | 32.2 | 84.72 | 55.3 | 67.34 | 60.77 | 65.95 | 63.68 |
| Best among experts | 71.33 | 91.71 | 69.44 | 64.57 | **82.5** | 61.55 | 33.2 | 86.6 | 56.91 | 70.38 | 64.47 | 72.03 | 68.72 |
| MoE-Top-2 | 73.22 | 91.66 | 71.19 | 66.61 | 81.45 | 64.11 | 34.7 | 87.22 | 58.61 | 72.98 | 66.52 | 72.63 | 70.08 |
| MoE-Top-4 | **73.48** | **91.85** | **71.36** | **67.33** | 81.74 | **64.52** | **35.05** | **87.33** | **58.78** | **73.31** | **66.95** | **73.07** | **70.40** |

Table 18: Zero-shot classification performance on the 13.6M dataset.

| Dataset | COCO IR | COCO TR | Urban IR | Urban TR | DCI IR | DCI TR | Flickr 8K IR | Flickr 8K TR | Flickr 30K IR | Flickr 30K TR | Avg. |
|---|---|---|---|---|---|---|---|---|---|---|---|
| Worst among experts | 42.83 | 54.58 | 85.8 | 86.6 | 54.38 | 41.82 | 66.98 | 78.0 | 70.94 | 85.3 | 66.72 |
| Best among experts | 46.87 | 59.72 | 92.3 | 92.8 | 59.49 | 53.62 | 71.76 | 82.1 | 76.48 | 89.8 | 72.49 |
| MoE-Top-2 | 48.17 | 62.84 | 93.4 | **93.4** | 59.54 | 53.98 | **72.72** | 83.8 | 76.86 | 91.3 | 73.6 |
| MoE-Top-4 | **48.28** | **63.02** | **93.6** | **93.4** | **59.67** | **54.17** | 72.70 | **84.6** | **76.96** | **91.6** | **73.8** |

Table 19: Zero-shot retrieval performance on the 13.6M dataset.

Table 4 in the main text presents results from the model trained at the smallest scale using the ShareGPT4V dataset with a ViT-B/16 backbone, which may not fully capture expert collaboration. Additionally, due to computational constraints, we employed only TopK=2 with 4 experts in this experiment. Since each expert specializes in different feature domains, the limited TopK and small number of experts (2 and 4) restrict the number of possible collaborations and may lead to one expert dominating, which aligns with the reviewer's observation. To better examine expert collaboration, we decided to scale up the number of experts to 8 and train the CLIP ViT-L/14 model on the "Merged Dataset" consisting of ShareGPT4V, LLaVA-ReCap-CC3M, and LLaVA-ReCap-CC12M (Total: 13.6M images). To optimize computational resources, we set TopK=2 during training, and used TopK=2 and TopK=4 during inference. We compare the worst and best performance of individual experts across each dataset with the corresponding performance of the MoE model. The results are presented in the two tables below (top-1 accuracy and retrieval).

These results in Tables 18 and 19 demonstrate that with a larger number of total and activated experts, the unified model benefits from a more diverse combination of expert knowledge, leading to a clearer gap in performance between the unified model and individual experts in both classification and retrieval tasks. We also observe that using a larger TopK during inference can improve overall performance, even when a smaller TopK is used during training. We hypothesize that training with a larger TopK, such as TopK=4, could further narrow the gap and lead to even better results.

## H    DETAILS OF LARGE LANGUAGE MODELS USAGE

We employ Large Language Model (LLMs) solely to assist with polishing the writing of this manuscript. The LLMs are used to improve grammar, clarity, and style in certain sections, but they do not contribute to the conceptual development, experimental design, data analysis, or interpretation of results. All scientific content, ideas, and conclusions are derived entirely by the human authors.

