# OpenReview forum: "CLIP-FMoE: Scalable CLIP via Fused Mixture-of-Experts with Enforced Specialization"
_ICLR.cc/2026/Conference — ICLR 2026 Poster_

### Official Review · Reviewer_jDHU · 2025-10-27

**Soundness:** 2
**Presentation:** 4
**Contribution:** 3
**Rating:** 6
**Confidence:** 2

**Summary:**

This paper proposes CLIP-FMoE, a mixture-of-experts scaling method for CLIP models. CLIP-FMoE combines an Isolated Constrained Contrastive Learning stage and a fused MoE training stage to preserve pretrained knowledge. Experiments on zero-shot classification, image-text retrieval, and long-caption benchmarks show modest improvements over existing baselines.

**The assigned paper is outside my area of expertise. My reviews are based on my limited understanding.**

**Strengths:**

1. This paper addresses an important challenge and is valuable for multimodal foundation model development.
2. The method is evaluated on diverse benchmarks, including standard and fine-grained classification, retrieval tasks, and long-context understanding.
3. The paper is clearly written and well-organized.

**Weaknesses:**

1. It seems like novelty is incremental: Both ICCL and Fusion Gate largely reuse familiar concepts (e.g., semantic clustering of data, constrained batch sampling, and gating-based interpolation).
2. The authors should provide stronger evidence for expert specialization. The current evaluation of expert behavior relies only on limited task-wise performance comparisons (Table 4) and does not convincingly show that semantic clustering leads to meaningful routing or complementary feature learning.

**Questions:**

Please refer to Weaknesses.

---

> ### Author Response · Authors · 2025-11-22
> **Thanks for constructive feedback**
>
> We want to thank the reviewer for positive evaluation and valuable feedback for improving the paper. We are pleased to address your remaining concerns below.
> ### **R4-W1: It seems like novelty is incremental: Both ICCL and Fusion Gate largely reuse familiar concepts (e.g., semantic clustering of data, constrained batch sampling, and gating-based interpolation).**
>
> We want to gently direct the reviewer's attention to our responses to R3-W1 and R3-W4 (Reviewer 2spG), where we highlight the unique value of our approach compared with prior work in model merging, dynamic network composition, and continual-learning methods for catastrophic forgetting. Our contribution lies not in any single component but in a practical recipe that simultaneously achieves two goals. First, it preserves and leverages pretrained CLIP knowledge while efficiently adapting to new data under limited computational resources. Second, it enforces expert specialization by training experts on semantically clustered data and mitigating token-imbalance effects in MoE through the train-before-unify pipeline. To achieve these goals, we intentionally build this framework from simple, well-understood, and effective ingredients, as also noted by Reviewer sfPq, to provide a robust and reproducible recipe that opens up new directions, for example, extensions to non-CLIP transformer models (see R2-Q2, response to Reviewer Wka5), and a complementary baseline for future work to investigate.
>
> ### **R4-W2: The authors should provide stronger evidence for expert specialization. The current evaluation of expert behavior relies only on limited task-wise performance comparisons (Table 4) and does not convincingly show that semantic clustering leads to meaningful routing or complementary feature learning.**
>
> To address this concern, we conduct an additional analysis on the MMVP benchmark [1] using the same experts (ViT-B/16) as in Table 4 and show that they capture diverse and complementary attributes. MMVP evaluates a CLIP model by asking it to select the correct image, given a textual statement, from a pair of visually similar images. The evaluation set is carefully categorized by human annotators into nine fine-grained visual attributes.
>
> The results below indicate clear attribute-level specialization across experts. For example, Expert 0 performs best on Quantity and Count. Expert 1 excels in Orientation and Direction and Presence of Specific Features. Expert 2 focuses on Color and Appearance and Structural Characteristics, while Expert 3 is strongest on State and Condition and shares the lead on Texts. This pattern supports our claim that semantic clustering induces meaningful routing and complementary feature learning rather than redundant experts.
>
> | Attribute | Expert 0 | Expert 1 | Expert 2 | Expert 3 |
> |---|---|---|---|---|
> | Orientation and Direction | 13.33 | **20** | 0 | 6.67 |
> | Presence of Specific Features | 6.67 | **20** | 13.33 | 6.67 |
> | State and Condition | 40 | 40 | 40 | **46.67** |
> | Quantity and Count | **20** | 13.33 | 13.33 | 13.33 |
> | Positional and Relational Context | 0 | 0 | **20** | 6.67 |
> | Color and Appearance | 53.33 | 66.67 | **73.33** | 66.67 |
> | Structural Characteristics | 20 | 26.67 | **33.33** | 26.67 |
> | Texts | 6.67 | 6.67 | **13.33** | **13.33** |
> | Viewpoint and Perspective | 13.33 | 6.67 | **20** | 6.67 |
>
> [1] Tong, Liu et al. "Eyes Wide Shut? Exploring the Visual Shortcomings of Multimodal LLMs", *CVPR*, 2024

---

> > ### Comment · Reviewer_jDHU · 2025-11-27
> >
> > Thanks for the additional experiments on expert specialization. I have no further questions. The new results address my earlier concerns, and I will keep my score as *marginally above the acceptance threshold.*
> >
> > **I would also like to remind the AC that this paper is outside my main area of expertise, and my evaluation is based on a limited understanding.** Therefore, my review should be taken as only one part of the overall assessment. I also appreciate the AC’s work in coordinating the review process.

---

> > > ### Author Response · Authors · 2025-12-02
> > >
> > > Thank you for revisiting our work and confirming that the new experiments address your earlier concerns. We appreciate your note that this paper is slightly outside your main area of expertise, and your questions still helped us improve the paper.

---

### Official Review · Reviewer_2spG · 2025-10-28

**Soundness:** 2
**Presentation:** 3
**Contribution:** 2
**Rating:** 4
**Confidence:** 3

**Summary:**

This paper presents CLIP-FMoE, a scalable MoE framework for CLIP to handle diverse visual-text data efficiently. The key idea is a two-stage clustering mechanism that partitions data by visual similarity, enabling each expert to specialize in a subset of the modality space. A lightweight Fusion Gate is introduced to adaptively aggregate expert outputs, balancing specialization and generalization. The experiments demonstrate improvements in zero-shot and retrieval benchmarks over several baselines.

**Strengths:**

1. The paper is well-written and the proposed method is presented clearly, making it easy to follow.
2. Experimental results are consistent across multiple benchmarks, showing solid improvements over vanilla CLIP and other MoE-based baselines.

**Weaknesses:**

1. The novelty of the fusion gate is somewhat incremental, as gating mechanisms for interpolating between network modules are a common way in deep learning. The paper could strengthen its contribution by better positioning this component with respect to prior art in model merging or dynamic network composition.
2. The analysis of expert collaboration is limited. The expert specialization results in Table 4 suggest the unified model often matches the performance of the single best expert, rather than demonstrating strong synergistic gains from combining multiple experts.
3. The sensitivity of balancing loss coefficient alpha in Eq. 5 has not been evaluated. The authors use a fixed value of 0.01, but an ablation study would be necessary to understand its impact and the robustness of the model to this choice.
4. In the context of preventing catastrophic forgetting, continual learning literature offers several relevant strategies. Besides MoE-based approaches such as MoE-Adapters [1] and L2P [2], there also exist regularization-based approaches e.g. ZSCL [3], orthogonality-based approaches e.g. PGP [4] and feature compression approaches e.g. LADA [5]. These methods should be discussed in the related work section.



[1] Yu, Jiazuo, et al. "Boosting continual learning of vision-language models via mixture-of-experts adapters." *Proceedings of the IEEE/CVF Conference on Computer Vision and Pattern Recognition*. 2024.

[2] Wang, Zifeng, et al. "Learning to prompt for continual learning." *Proceedings of the IEEE/CVF conference on computer vision and pattern recognition*. 2022.

[3] Zheng, Zangwei, et al. "Preventing zero-shot transfer degradation in continual learning of vision-language models." *Proceedings of the IEEE/CVF international conference on computer vision*. 2023.

[4] Qiao, Jingyang, et al. "Prompt gradient projection for continual learning." *The Twelfth International Conference on Learning Representations*. 2024.

[5] Luo, Mao-Lin, et al. "LADA: Scalable Label-Specific CLIP Adapter for Continual Learning." *Forty-second International Conference on Machine Learning*.

**Questions:**

1. Could the authors provide details on the sample distribution across the generated data clusters? Specifically, how balanced were the clusters?
2. In Stage 2, only the router and fusion gate are trained. Have the authors experimented with also fine-tuning the expert weights with a small learning rate? It seems plausible that allowing for minor adjustments during this unification stage could further improve performance.

---

> ### Author Response · Authors · 2025-11-22
> **Thanks for many insightful questions**
>
> We would like to thank the reviewer for many insightful questions and feedback. We are pleased to address your remaining concerns below.
>
> ### **R3-W1: The novelty of the fusion gate is somewhat incremental, as gating mechanisms for interpolating between network modules are a common way in deep learning. The paper could strengthen its contribution by better positioning this component with respect to prior art in model merging or dynamic network composition.**
>
> We highly appreciate the suggestions about related literature. We will integrate the following discussion into the Related Work section, comparing our approach to model merging and dynamic network composition.
>
> The problem of combining pre-trained models with task-specific knowledge has been explored through various paradigms. Model merging techniques such as task arithmetic [1] enable linear combinations of model weights to compose capabilities, while LiNeS [2] introduces post-training layer scaling to prevent catastrophic forgetting during merging. Temporal merging approaches [3] address how to combine multimodal models trained at different time points, and recent work on task arithmetic in tangent space [4] demonstrates improved editing of pre-trained models through geometric considerations. Beyond static merging, dynamic composition methods have gained attention: mixture-of-experts architectures [5] employ gating mechanisms to route inputs across specialized modules, while Model Soups [6] demonstrate that averaging weights of multiple fine-tuned models improves accuracy.
>
> These approaches typically merge models post-hoc [1,6] or interpolate between fixed components. In contrast, our fusion gate mechanism is integrated directly into the training pipeline at two critical stages: (1) during expert learning in Stage-1, where each expert blends signals from the frozen base model through a fused dense layer, and (2) during Stage-2, where we fuse the base path with the entire MoE mixture rather than just interpolating among experts. This dual-stage design addresses distinct objectives—Stage-1 gating preserves zero-shot capabilities while Stage-2 gating enhances retrieval diversity—as demonstrated by our ablation studies showing that both stages are necessary for optimal performance. Unlike prior merging methods that operate on pre-trained components, our approach learns the fusion dynamically and hierarchically throughout the training process.
>
> [1] Ilharco et al., Editing Models with Task Arithmetic, *ICLR*, 2023.
>
> [2] Wang et al., LiNeS: Post-training Layer Scaling Prevents Forgetting and Enhances Model Merging, *ICLR*, 2025.
>
> [3] Dziadzio et al., How to Merge Your Multimodal Models Over Time?, *CVPR*, 2025.
>
> [4] Ortiz-Jimenez et al., Task Arithmetic in the Tangent Space: Improved Editing of Pre-Trained Models, *NeurIPS*, 2023.
>
> [5] Fedus et al., Switch Transformers: Scaling to Trillion Parameter Models with Simple and Efficient Sparsity, *JMLR*, 2022.
>
> [6] Wortsman et al., Model soups: averaging weights of multiple fine-tuned models improves accuracy without increasing inference time, *ICML*, 2022.

---

> ### Author Response · Authors · 2025-11-22
>
> ### **R3-W2: The analysis of expert collaboration is limited. The expert specialization results in Table 4 suggest the unified model often matches the performance of the single best expert, rather than demonstrating strong synergistic gains from combining multiple experts.**
>
> We believe that Table 4 in the main text presents results from the model trained at the smallest scale using the ShareGPT4V dataset with a ViT-B/16 backbone, which may not fully capture expert collaboration. Additionally, due to computational constraints, we employed only TopK=2 with 4 experts in this experiment. Since each expert specializes in different feature domains, the limited TopK and small number of experts (2 and 4) restrict the number of possible collaborations and may lead to one expert dominating, which aligns with the reviewer's observation. To better examine expert collaboration, we decided to scale up the number of experts to 8 and train the CLIP ViT-L/14 model on the "Merged Dataset" consisting of ShareGPT4V, LLaVA-ReCap-CC3M, and LLaVA-ReCap-CC12M (Total: 13.6M images). To optimize computational resources, we set TopK=2 during training, and used TopK=2 and TopK=4 during inference. We compare the worst and best performance of individual experts across each dataset with the corresponding performance of the MoE model. The results are presented in the two tables below (top-1 accuracy and retrieval).
>
> These results demonstrate that with a larger number of total and activated experts, the unified model benefits from a more diverse combination of expert knowledge, leading to a clearer gap in performance between the unified model and individual experts in both classification and retrieval tasks. We also observe that using a larger TopK during inference can improve overall performance, even when a smaller TopK is used during training. We hypothesize that training with a larger TopK, such as TopK=4, could further narrow the gap and lead to even better results.
>
> | Dataset | Cars | Food | Sun | Eurosat | Cal | IN-a | IN-o | IN-r | IN-s | IN-1K | IN-v2 | FLow | Avg. |
> | :--- | :---: | :---: | :---: | :---: | :---: | :---: | :---: | :---: | :---: | :---: | :---: | :---: | :---: |
> | Worst among experts | 65.69 | 88.49 | 64.71 | 48.83 | 75.22 | 54.88 | 32.2 | 84.72 | 55.3 | 67.34 | 60.77 | 65.95 | 63.68 |
> | Best among experts | 71.33 | 91.71 | 69.44 | 64.57 | **82.5** | 61.55 | 33.2 | 86.6 | 56.91 | 70.38 | 64.47 | 72.03 | 68.72 |
> | MoE-Top-2 | 73.22 | 91.66 | 71.19 | 66.61 | 81.45 | 64.11 | 34.7 | 87.22 | 58.61 | 72.98 | 66.52 | 72.63 | 70.08 |
> | MoE-Top-4 | **73.48** | **91.85** | **71.36** | **67.33** | 81.74 | **64.52** | **35.05** | **87.33** | **58.78** | **73.31** | **66.95** | **73.07** | **70.40** |
>
> | Dataset | COCO IR | COCO TR | Urban IR | Urban TR | DCI IR | DCI TR | Flickr 8K IR | Flickr 8K TR | Flickr 30K IR | Flickr 30K TR | Avg. |
> | :--- | :---: | :---: | :---: | :---: | :---: | :---: | :---: | :---: | :---: | :---: | :---: |
> | Worst among experts | 42.83 | 54.58 | 85.8 | 86.6 | 54.38 | 41.82 | 66.98 | 78 | 70.94 | 85.3 | 66.72 |
> | Best among experts | 46.87 | 59.72 | 92.3 | 92.8 | 59.49 | 53.62 | 71.76 | 82.1 | 76.48 | 89.8 | 72.49 |
> | MoE-Top-2 | 48.17 | 62.84 | 93.4 | **93.4** | 59.54 | 53.98 | **72.72** | 83.8 | 76.86 | 91.3 | 73.6 |
> | MoE-Top-4 | **48.28** | **63.02**| **93.6** | **93.4** | **59.67** | **54.17** | 72.7 | **84.6** | **76.96** | **91.6** | **73.8** |

---

> ### Author Response · Authors · 2025-11-22
>
> ### **R3-W3: The sensitivity of balancing loss coefficient alpha in Eq. 5 has not been evaluated. The authors use a fixed value of 0.01, but an ablation study would be necessary to understand its impact and the robustness of the model to this choice.**
>
> Since the use of a balancing loss is common when training MoE models, and to ensure a fair comparison, we set a fixed value of 0.01 as suggested by CLIP-MoE [1] without further exploration. We acknowledge the importance of evaluating the sensitivity of the balancing loss coefficient alpha, as raised by the reviewer. To address this, we re-trained Stage 2 of the CLIP-FMoE experiments (as shown in Table 1 of the main text) with different values of alpha. The results show no significant differences in performance across the cases, suggesting that the proposed method is robust to variations in alpha. We hypothesize that alpha may have a greater impact on token balancing when using a model with a larger number of experts, or that our train-before-unify mechanism may already partially address this issue. We plan to investigate this further in future work.
>
>
> | Alpha | Cars | Coun | FGVC | Food | Sun | Cal | Euro | Flow | Pets | IN-1K | IN-v2 | Avg. |
> | --- | --- | --- | --- | --- |---| --- | --- | --- | --- | --- | --- | --- |
> | 0 | 74.47 | 28.02 | 29.04 | 92.3 | 71.87 | 81.76 | 68.24 | 73.48 | 91.44 | 74.85 | 68.64 | 68.56 |
> | 0.01 (default) | 74.62 | 28.02 | 29.58 | 92.32 | 71.86 | 81.79 | 68.31 | 73.54 | 91.58 | 74.87 | 68.66 | 68.65 |
> | 0.05 | 74.59 | 28.0 | 29.37 | 92.32 | 71.88 | 81.81 | 68.28 | 73.57 | 91.52 | 74.86 | 68.67 | 68.62 |
> | 0.1 | 74.59 | 28.01 | 29.58 | 92.32 | 71.88 | 81.82 | 68.24 | 73.52 | 91.55 | 74.86 | 68.61 | 68.63 |
>
> | Alpha | COCO IR | COCO TR | Flickr 8K IR | Flickr 8K TR | Flickr 30K IR | Flickr 30K TR | Avg. |
> | --- | --- | --- | --- | --- | --- | --- | --- |
> | 0 | 49.61 | 66.56 | 73.68 | 85.9 | 78.28 | 92.6 | 74.44 |
> | 0.01 (default) | 49.6 | 66.66 | 73.68 | 86.1 | 78.42 | 92.3 | 74.46 |
> | 0.05 | 49.59 | 66.6 | 73.68 | 86.1 | 78.42 | 92.3 | 74.45 |
> | 0.1 | 49.58 | 66.6 | 73.68 | 86 | 78.4 | 92.3 | 74.43 |
>
> [1] CLIP-MoE: Towards Building Mixture of Experts for CLIP with Diversified Multiplet Upcycling, *arXiv*, 2024
>
> ### **R3-W4: In the context of preventing catastrophic forgetting, continual learning literature offers several relevant strategies. Besides MoE-based approaches such as MoE-Adapters [1] and L2P [2], there also exist regularization-based approaches e.g. ZSCL [3], orthogonality-based approaches e.g. PGP [4] and feature compression approaches e.g. LADA [5]. These methods should be discussed in the related work section.**
> Thank you for poiting out the missing related continual learning literature. We will add the following dedicated paragraph about preventing catastrophic forgetting into the related works section.
>
> To mitigate catastrophic forgetting, the continual learning literature has explored a diverse set of strategies. Significant attention has been directed toward Mixture-of-Experts (MoE) and prompt-based frameworks, such as MoE-Adapters [1] and L2P [2], which dynamically adapt model architecture or input prompts. Parallel efforts have focused on regularization-based techniques, notably ZSCL [3], to constrain weight updates, while others have employed orthogonality-based constraints like PGP [4] to minimize interference between tasks. Additionally, feature compression methods such as LADA [5] have been proposed to enhance scalability. Distinguishing itself from these existing paradigms, our work introduces a fusion gate mechanism, designed to *explicitly* preserve pretrained knowledge during the learning process while allowing experts to acquire new complementary representations.

---

> ### Author Response · Authors · 2025-11-22
>
> ### **R3-Q1: Could the authors provide details on the sample distribution across the generated data clusters? Specifically, how balanced were the clusters?**
> We use standard K-means clustering, implemented by the Faiss library [1], to form the clusters. The distribution of clusters (used for the experiments in Table 1) is shown in the table below, demonstrating that the clusters are not significantly imbalanced. During training, we aim for parallel training of each expert with similar time allocation, which necessitates reasonably balanced clusters. While balance cannot be strictly guaranteed during the clustering process, the number of clusters (N) is relatively small compared to the total number of samples, which generally ensures that the clusters remain balanced.
>
> | Cluster 1 | Cluster 2 | Cluster 3 | Cluster 4 |
> |:--:|:--:|:-:|:--:|
> |0.26|0.28|0.28|0.19|
>
> [1] Jeff Johnson et al. "Billion-scale similarity search with GPUs", *IEEE Transactions on Big Data*, 2019
>
> ### **R3-Q2: In Stage 2, only the router and fusion gate are trained. Have the authors experimented with also fine-tuning the expert weights with a small learning rate? It seems plausible that allowing for minor adjustments during this unification stage could further improve performance.**
>
> We appreciate the suggestion, and we conducted additional experiments to evaluate a Stage-2 variant where we **fine-tune the experts instead of freezing them**. Starting from the same experts trained in Stage 1 and using the same experimental setup as in Table 1 of the paper, we perform Stage-2 unification with all experts made trainable. In the standard Stage 2, we train only the router (learning rate (5e-5); in the fine-tuning variant, we additionally update the expert weights with a smaller learning rate of (1e-5). The results are reported in the two tables below. Although this fine-tuning variant is approximately **2.7× more expensive** than the standard setting, it actually **degrades zero-shot classification performance and slightly worsens retrieval accuracy**, confirming that freezing experts in Stage 2 is the preferable choice.
>
>
> | Model|Cars|Coun|FGVC|Food|Sun|Cal|Euro|Flow|Pets|IN-1K| IN-v2  | Avg.|
> |:-|:-:|:-:|:-:|:-:|:-:|:-:|:-:|:-:|:-:|:-:|:-:|:-:|
> |CLIP Base| 77.93  | 31.88 | 31.77 | 93.07  | 67.56  | 83.27| 62.59  | 79.2| 93.21  | 75.54  | 69.84  | 69.62  |
> ||
> | **Freeze Experts**| **74.62**  | **28.02**| **29.58**| **92.32**  | **71.86**  | **81.79**  | 68.31  | **73.54** | **91.50**| **74.87**  | **68.66**  | **68.65**  |
> | **Fine-tune Experts** | 68.98| 24.88 | 27.09 | 90.94  | 71.04  | 79.46  | **69.31**  | 68.71  | 90.02| 73.07  | 66.74  | 66.39  |
>
> |Model| COCO IR R@1 |COCO IR R@5 |COCO TR R@1|COCO TR R@5 |Flickr 8K IR R@1|Flickr 8K IR R@5 |Flickr 8K TR R@1 | Flickr 8K TR R@5 | Flickr 30K IR R@1 | Flickr 30K IR R@5 | Flickr 30K TR R@1 |Flickr 30K TR R@5 |Avg.|
> |:---|:-:| :---: |:---:| :---:| :---: | :---: | :---: | :---: | :---: | :---: | :---: | :---: | :---: |
> |CLIP Base|36.51|61.01|56.38|79.36|61.60|86.36 |77.40|94.00|65.22|87.26|85.10|97.30|73.96|
> ||
> | **Freeze Experts** | **49.60** | **73.69** | **66.66** | **86.84**|**73.68**| **93.40** | **86.10** | **97.40** | **78.42**|**94.58**| **92.30** | **99.50** | **82.68** |
> | **Fine-tune Experts** | 49.17 | 73.60 | 63.86 | 84.34 | 73.12 |93.30|83.40| 97.00 | 78.34 | 95.10 | 91.30 | 99.20 | 81.81 |

---

> > ### Comment · Reviewer_2spG · 2025-11-23
> >
> > I would like to thank the authors for their rebuttal. The additional experiments provided in the response are sufficient and have effectively resolved my previous questions.
> >
> > It is an interesting finding that fine-tuning the expert weights alongside the router and fusion gate in the second stage leads to a performance drop on many datasets compared to training only the router and fusion gate. I suggest incorporating these experimental results and the related work on preventing catastrophic forgetting into the final paper.
> >
> > A minor note: there appears to be a small mistake in the reported table for Flickr 30K IR R@5; the bolded value should be 95.10, not 94.58.
> >
> > I have decided to raise my score.

---

> > > ### Author Response · Authors · 2025-11-24
> > >
> > > We sincerely appreciate your thoughtful reconsideration of the score and your careful note about the missing boldface value; we will incorporate these results and the correction into the final version of the paper.

---

### Official Review · Reviewer_Wka5 · 2025-10-30

**Soundness:** 3
**Presentation:** 3
**Contribution:** 2
**Rating:** 6
**Confidence:** 3

**Summary:**

This paper addresses the limitations of existing MoE adaptations for CLIP (high sequential training overhead and degraded zero-shot capabilities) by proposing CLIP-FMoE, an approach integrating MoE into CLIP fine-tuning via two core designs: Isolated Constrained Contrastive Learning (ICCL), a two-stage pipeline that performs two-level semantic clustering to train specialized experts in parallel and reduce token imbalance in MoE, and a Fusion Gate mechanism to mitigate catastrophic forgetting of pre-trained knowledge ; extensive experiments on benchmarks (11 zero-shot classification datasets, standard/fine-grained/long-context retrieval tasks) show CLIP-FMoE preserves zero-shot capabilities (average accuracy drop <1% vs. original CLIP), outperforms baselines like Vanilla Fine-tuning and CLIP-MoE, and maintains efficiency (72.5% less training time than CLIP-MoE, 14% higher inference FLOPs) ; its main contributions include introducing CLIP-FMoE with ICCL and Fusion Gate, ensuring expert specialization, and demonstrating robust performance across context lengths and dataset sizes .

**Strengths:**

1. **Clear Motivation**: The paper designs a two-stage training pipeline (Isolated Constrained Contrastive Learning, ICCL) and a Fusion Gate mechanism to address key issues in CLIP scaling: it uses two-level semantic clustering (coarse-grained primary clusters + fine-grained sub-clusters) for parallel expert training in Stage 1, ensuring experts learn non-overlapping domain knowledge and avoiding the accumulated errors of sequential training in prior CLIP-MoE; the Fusion Gate dynamically balances pre-trained CLIP knowledge and expert-specific expertise in both training stages, effectively mitigating catastrophic forgetting.



2. **Solid Experimental Design**:  The experiments cover diverse task scenarios: 11 zero-shot classification benchmarks (including fine-grained, scene, and general datasets), 6 retrieval tasks (standard MS COCO, fine-grained DOCCI, long-context Urban-1K), and long-context understanding (extending context length from 77 to 248 tokens). All methods use the same CLIP ViT-L/14 backbone and fine-tune only odd layers in the second half of the encoder, ensuring fair comparison detailed hyperparameters (e.g., batch size, learning rate) for different datasets (CC3M, ShareGPT4V) and extended experimental results (e.g., 17 classification benchmarks in the appendix) are provided, enhancing result reproducibility.

 3. **High computational efficiency**: CLIP-FMoE achieves an excellent trade-off between performance and computational cost: compared with CLIP-MoE, it reduces total training time  through parallel expert training; its inference FLOPs only increase by 14%, while the peak trainable parameters are 80% fewer than Up-cycling, reducing GPU memory pressure. Additionally, it maintains robustness across scenarios—performing well in both resource-constrained environments (due to low memory usage) and diverse practical tasks (e.g., fine-grained product recognition, long-text image retrieval), expanding the application scope of CLIP-style models.

**Weaknesses:**

1. **Incremental Contribution**: CLIP-FMoE’s core components are all combinations of mature existing technologies in the field: the MoE architecture , contrastive learning with InfoNCE loss , K-means semantic clustering, and the Fusion Gate —no breakthrough technical ideas  and insights are proposed. In terms of performance, its improvement is only incremental: in zero-shot classification tasks (Table 1), its average accuracy (68.65) is only 2.18 percentage points higher than the second-best baseline CLIP-MoE (66.47), and even 1.19 percentage points lower than the original CLIP (69.84); in retrieval tasks (Table 2), the average R@1 is only 0.39 percentage points higher than CLIP-MoE, with a tiny improvement margin that fails to open up new technical directions.

**Questions:**

# Questions for the Authors
Given that this paper presents an engineering-sound work on CLIP scaling via CLIP-FMoE, i have two questions:
1. What new insights do you believe this work provides to the field of CLIP scaling and MoE adaptation for vision-language models, which were not sufficiently addressed or recognized in prior studies?
2. The current framework is tailored for CLIP fine-tuning. Do you think the core design (e.g., ICCL’s two-level clustering, Fusion Gate) can be extended to train non-CLIP models? If so, what key adjustments might be needed; if not, what inherent limitations of the framework restrict such extension?

---

> ### Author Response · Authors · 2025-11-22
> **Thanks for valuable feedback**
>
> We want to thank the reviewer for positive evaluation and valuable feedback for improving the paper. We address your remaining concerns below.
>
> ### **R2-W1/Q1: What new insights do you believe this work provides to the field of CLIP scaling and MoE adaptation for vision-language models, which were not sufficiently addressed or recognized in prior studies?**
>
> We thank the reviewer for raising this insightful questions. First, we want to gently direct your attention to our responses to R3-W1 and R3-W4 (Reviewer 2spG), where we highlight the unique value of our approach compared with prior work in model merging, dynamic network composition, and continual-learning methods for catastrophic forgetting. Our contribution lies not in any single component but in a practical recipe that simultaneously achieves two goals. First, it preserves and leverages pretrained CLIP knowledge while efficiently adapting to new data under limited computational resources. Second, it enforces expert specialization by training experts on semantically clustered data and mitigating token-imbalance effects in MoE through the train-before-unify pipeline. To achieve these goals, we intentionally build this framework from simple, well-understood, and effective ingredients, as also noted by Reviewer sfPq, to provide a robust and reproducible recipe that opens up new directions, for example, extensions to non-CLIP transformer models as you suggest in Question 2, and a complementary baseline for future work to build upon.
>
> ### **R2-Q2: The current framework is tailored for CLIP fine-tuning. Do you think the core design (e.g., ICCL’s two-level clustering, Fusion Gate) can be extended to train non-CLIP models? If so, what key adjustments might be needed; if not, what inherent limitations of the framework restrict such extension?**
>
> We believe that the core design of our framework, based on the traditional MoE architecture applied to the MLP layer of transformer blocks, is applicable to any other transformer-based models commonly used in state-of-the-art architectures. Therefore, CLIP-FMoE can be extended to non-CLIP models, particularly transformer-based ones. For example, recent work such as MoE-LLaVA [1] demonstrates the scalability of LLaVA using the Up-cycling MoE method [2] to achieve improved performance. CLIP-FMoE could similarly be applied to LLaVA by training each expert separately and then unifying them into the MoE-LLaVA model, replacing the base-weight replicated MLPs with specialized experts. The Fusion Gate mechanism from CLIP-FMoE can be seamlessly integrated into the LLaVA transformer block during the unification stage (Stage 2), as originally proposed.
>
> However, some adjustments are necessary to adapt ICCL's two-level clustering for LLaVA. Unlike multimodal contrastive learning, multimodal generative models like LLaVA require triplets of image, prompt input, and text output. A naive clustering approach could involve clustering each data modality separately, and then using the Cartesian product to define the joint clusters. However, for feature extraction in the clustering process, using LLaVA itself like CLIP may not be cost-efficient. Instead, employing lighter-weight, more powerful embedding models could be more suitable. Another consideration is the use of data filtering methods to select more quality samples, as discussed in our response to R1-W3 (reviewer sfPq). This would help improve the base model’s performance, such as enhancing visual reasoning capabilities in LLaVA.
>
> In summary, the Fusion Gate and ICCL mechanisms can be extended to any models eploying the vanilla MoE architecture, and even non-transformer models like MoE-CNN [3] or MoE-Mamba [4], provided a suitable clustering mechanism is chosen. We appreciate the reviewer’s constructive question and view extending our method to non-CLIP models as a promising future direction.
>
> [1] Lin, Tang et al. "MoE-LLaVA: Mixture of Experts for Large Vision-Language Models", *IEEE Transactions on Multimedia*, 2025
>
> [2] Komatsuzaki, Puigcerver et al. "Sparse Upcycling: Training Mixture-of-Experts from Dense Checkpoints", *ICLR*, 2023
>
> [3] Yang, Cai et al. "Robust Mixture-of-Expert Training for Convolutional Neural Networks", *ICCV*, 2023
>
> [4] Pióro, Ciebiera et al. "MoE-Mamba: Efficient Selective State Space Models with Mixture of Experts", *ICLR 2024 Workshop on Understanding of Foundation Models*, 2024

---

> > ### Comment · Reviewer_Wka5 · 2025-11-28
> >
> > Thank you for your thoughtful response addressing the raised concerns. Your explanations have effectively clarified the core contributions of the work and its extensibility, which reinforces my positive evaluation of this paper.
> >
> > Regarding the new insights into CLIP scaling and MoE adaptation, your emphasis on the "practical recipe"—integrating existing components to balance pretrained knowledge preservation, resource efficiency, and expert specialization—is well-reasoned. The train-before-unify pipeline’s role in mitigating token imbalance in MoE, coupled with the framework’s robustness and reproducibility, indeed fills a gap in prior fragmented studies on model merging and continual learning.
> >
> > For the extensibility to non-CLIP models, your analysis of adapting ICCL clustering and Fusion Gate to transformer-based models (e.g., LLaVA) is convincing. The proposed adjustments—such as multimodal joint clustering and lightweight embedding models for cost efficiency—demonstrate a clear understanding of cross-model adaptation challenges. Extending to MoE-CNN or MoE-Mamba also opens promising directions, as you noted.
> >
> > In summary, your response sufficiently addresses my questions. I maintain my positive score and reaffirm my support for the paper’s acceptance.

---

> > > ### Author Response · Authors · 2025-12-02
> > >
> > > Thank you for your detailed and encouraging assessment. Your continued positive score and clear support for acceptance are greatly appreciated and very motivating for us.

---

### Official Review · Reviewer_sfPq · 2025-11-01

**Soundness:** 3
**Presentation:** 4
**Contribution:** 3
**Rating:** 6
**Confidence:** 5

**Summary:**

This paper introduces CLIP-FMoE, a method for scaling Contrastive Language-Image Pre-training (CLIP) models using a Fused Mixture-of-Experts (MoE) architecture, with the aim to reduce computation cost when scaling. It involves two stages: Isolated Constrained Contrastive Learning (ICCL) where data is semantically clustered for separate experts training & specialization, and the integration of these experts into an MoE layer, by introducing the Fusion Gate mechanism to dynamically blend pre-trained knowledge with new, domain-specific expert knowledge. The method is evaluated on CC3M and achieves better performance consistently across tasks. Though, the method is not evaluated on billion-scale dataset training while the generalization of this method is expected to be further examined if more computational resource is available.

**Strengths:**

- The idea to use CLIP model to split the data via clustering is interesting.
- The performance is good and the result is convincing & promising.
- The fusing mechanism is indeed needed and this paper provides a good methodology to prevent forgetting and enable efficient scaling.
- Though not clearly stated in the method, but this paper consider shared and specialized parameters, which is good.

**Weaknesses:**

- **Ablation on Clustering Strategy**: The impact of the specific clustering algorithm and hyperparameters (e.g., choice of N and M clusters) on final performance is not thoroughly explored. It is unclear how sensitive the method is to the quality and granularity of the semantic clusters.
- **Single-modal clustering**: The paper uses image modality only for the clustering, but more discussion is needed. To fully discover the finegrained visual concept, the same image may need to be compared with different set of negative samples that focus on different aspects, thus it is unclear how the clustering over visual embeddings can help highlight the fine-grained visual elements. Also, as discussed in MoDE, the limited description in text is the primary cause of false negative. As such, more discussion is needed.
- **Generalization**: The method is only evaluated on small-scale CLIP training, and the generalization over much larger-scale dataset is hard to determine. I completely understand that the computation resources is a primary constraint, but I would still keen to know how your method can generalize.

**Questions:**

I am curious on how your approach generalize when trained on OpenAI CLIP ViT-Base. After all, the visual embedding quality of ViTs with different scale varies, which may hurt the performance.

Meanwhile, I am happy to increase my score further if the authors can address all my questions & weakness part properly.

---

> ### Author Response · Authors · 2025-11-22
> **Thanks for Constructive Feedback.**
>
> We want to thank the reviewer for positive evaluation and constructive feedback for improving the manuscript. We would like to address your remaining concerns below.
> ### **R1-W1: The impact of the specific clustering algorithm and hyperparameters (e.g., choice of N and M clusters) on final performance is not thoroughly explored. It is unclear how sensitive the method is to the quality and granularity of the semantic clusters.**
> We kindly refer the reviewer to Appendix F (Tables 15 and 16), where we explored the impact of the number of experts (N) and sub-clusters (M) on performance. Our results reveal a trade-off between classification and retrieval performance as N increases: while more experts improve zero-shot classification, they can slightly reduce retrieval accuracy. However, performance remains largely robust with respect to variations in M, with observed differences of less than 1%.
>
> ### **R1-W2: The paper uses image modality only for the clustering, but more discussion is needed. To fully discover the finegrained visual concept, the same image may need to be compared with different set of negative samples that focus on different aspects, thus it is unclear how the clustering over visual embeddings can help highlight the fine-grained visual elements. Also, as discussed in MoDE, the limited description in text is the primary cause of false negative. As such, more discussion is needed.**
>
> We want to gently direct the reviewer's attention to Appendix D, where we compared several clustering strategies: (i) random clustering, (ii) text-only clustering, (iii) image+text Cartesian product, (iv) image-only level-1 clustering, and (v) our standard two-level image clustering. Our standard two-level image clustering outperforms the others on average across both standard and fine-grained/long-caption retrieval tasks (see Tables 11–12). We hypothesize that this performance is due to the limited representation quality of CLIP's text encoder, which was trained on raw web-crawled captions rather than high-quality corpora generated by modern multimodal LLMs. The proposed clustering method not only leverages the image encoder for clustering, thereby avoiding the limitations of text-based clustering, but also ensures higher quality in-batch negatives through two-level clustering, which better captures fine-grained visual aspects.

---

> ### Author Response · Authors · 2025-11-22
>
> ### **R1-W3: Generalization: The method is only evaluated on small-scale CLIP training, and the generalization over much larger-scale dataset is hard to determine. I completely understand that the computation resources is a primary constraint, but I would still keen to know how your method can generalize.**
>
> We appreciate the reviewer’s constructive suggestion regarding the evaluation of our method on larger-scale datasets. To address this, we extended our experiments to a larger data regime (13.6M images) and compared it against smaller-scale settings.
>
> We introduced a "Merged Dataset" consisting of ShareGPT4V + LLaVA-ReCap-CC3M + LLaVA-ReCap-CC12M (Total: 13.6M images). We trained CLIP ViT-L/14 with an extended context length of 248 tokens. As shown in the tables below, CLIP-FMoE effectively generalizes to the larger dataset.
>
> | Data | Cars | Coun | FGVC | Food | Sun | Cal | Euro | Flow | Pets | IN-1K | IN-v2 | Avg. |
> | :--- | :---: | :---: | :---: | :---: | :---: | :---: | :---: | :---: | :---: | :---: | :---: | :---: |
> | OpenAI (400M) | 77.9 | 31.9 | 31.8 | 93.1 | 67.6 | 83.3 | 62.6 | 79.2 | 93.2 | 75.5 | 69.8 | 69.6 |
> |
> | ShareGPT4V (1.2M) | **75.8** | **30.1** | **29.8** | 92.4 | 65.1 | **84.4** | 67.2 | **76** | **93** | 73.9 | 67.8 | 68.7 |
> | LLaVA-ReCap-CC3M (2.8M) | 74.6 | 28 | 29.7 | **92.8** | **71.9** | 82.3 | **68** | 74.2 | 92.2 | **74.7** | **68.7** | **68.8** |
> | Merged Dataset (13.6 M) | 73.5 | 28.1 | 29.7 | 91.9 | 71.4 | 81.7 | 67.3 | 73.1 | 90.8 | 73.3 | 67.0 | 68.0 |
>
> | Data | COCO IR | COCO TR | DOCCI-Long IR | DOCCI-Long TR | IIW-Long IR | IIW-Long TR | Urban IR | Urban TR | DCI IR | DCI TR | Avg. |
> | :--- | :---: | :---: | :---: | :---: | :---: | :---: | :---: | :---: | :---: | :---: | :---: |
> | OpenAI (400M) | 36.5 | 56.4 | 63.3 | 66.0 | 90.5 | 90.3 | 55.8 | 68.3 | 33.9 | 36.0 | 59.7 |
> |
> | ShareGPT4V (1.2M) | 47.6 | 65.5 | 78.8 | 77.6 | 97.5 | **98.3** | 88.0 | 90.5 | 54.5 | 55.4 | 75.4 |
> | LLaVA-ReCap-CC3M (2.8M) | **49.5** | **67.0** | **84.9** | **81.9** | 97.8 | 96.8 | 93.4 | **94.0** | **59.9** | **58.1** | **78.3** |
> | Merged Dataset (13.6 M) | 48.3 | 63.0 | 83.8 | 80.3 | **98.5** | 95.6 | **93.6** | 93.4 | 59.7 | 54.2 | 77.0 |
>
> | Data | Experts (N)/TopK*/TopK** | Batchsize | # Captions |
> | :--- | :---: | :---: | :---: |
> | OpenAI (400M) | --None-- | 32768 | 1 |
> |
> | ShareGPT4V (1.2M) | 4/2/2 | 2048 | 4 |
> | LLaVA-ReCap-CC3M (2.8M) | 4/2/2 | 2048 | 4 |
> | Merged Dataset (13.6 M) | 8/2/4 | 1024 | 3 |
>
> Note: TopK values shown for MoE training (*) and inference (**).
> * Classification: The method effectively preserves pretrained knowledge in the 13.6M setting (Avg. 68.0%), comparable to the high-quality 1.2M and 2.8M baselines, and remains competitive with the OpenAI 400M zero-shot baseline.
> * Retrieval: All three fine-tuning settings significantly outperform the original CLIP. While the Merged Dataset (13.6M) performs slightly lower than the LLaVA-ReCap-CC3M (2.8M) setting on average, it achieves the best results on specific difficult benchmarks like IIW-Long (IR 98.5) and Urban (IR 93.6).
>
> Discussion on Scaling Constraints: We believe that the marginal performance drop in the Merged Dataset compared to the 2.8M setting is likely due to two experimental constraints rather than a limitation of the method itself:
> * **Batch Size Sensitivity**: Due to compute limits, the 13.6M model was trained with a batch size of 1024 (vs. 2048 for smaller datasets). Contrastive learning performance is heavily dependent on large batch sizes to provide sufficient negative samples [1, 2]. A smaller batch size on a larger, more diverse dataset likely hindered convergence.
> * **Data Quality vs. Quantity**: The Merged Dataset introduces CC12M, which contains significantly more noise than ShareGPT4V. Recent studies on Data Filtering Scaling Laws [3] and dataset robustness [4] demonstrate that simply mixing larger, noisier data sources does not guarantee better performance and can dilute the signal from high-quality captions.
>
> These results confirm that our method generalizes to larger scales, but optimal performance requires balancing batch size and data quality. A promising future direction could be focusing on incorporating advanced data filtering mechanisms and optimizing for smaller-batch training.
>
> [1] Chen, Zhang, et. al. "Why do We Need Large Batchsizes in Contrastive Learning? A Gradient-Bias Perspective.", *NeurIPS*, 2022
>
> [2] Chen, Kornblith, et al. "A Simple Framework for Contrastive Learning of Visual Representations", *ICML*, 2020
>
> [3] Gadre, Ilharco et al. "DATACOMP: In search of the next generation of multimodal datasets", *NeurIPS*, 2023
>
> [4] Nguyen, Ilharco, et al. "Quality Not Quantity: On the Interaction between Dataset Design and Robustness of CLIP", *NeurIPS*, 2022

---

> > ### Author Response · Authors · 2025-11-22
> >
> > ### **R1-Q: I am curious on how your approach generalize when trained on OpenAI CLIP ViT-Base. After all, the visual embedding quality of ViTs with different scale varies, which may hurt the performance.**
> >
> > We appreciate the reviewer’s suggestion to further investigate how our approach generalizes across different model scales. We want to gently point out that in most of our ablation studies, we use the OpenAI CLIP ViT-B/16, as shown in Tables 4, 11, 12, 15, and 16, where experiments are conducted with ViT-B/16 using a context length of 248 and trained on the ShareGPT4V dataset. To further investigate the generalization of CLIP-FMoE to other architectures, we provide a summary of the performance of it compared to OpenAI CLIP and Long-CLIP across multiple model scales. As shown in the tables below, while CLIP-FMoE performs well with the ViT-L/14 scale, we observe a slight degradation in performance when applying our method to the ViT-B/16, specifically in preserving the pretrained knowledge. However, for retrieval tasks, CLIP-FMoE still outperforms Long-CLIP, indicating that our approach remains competitive despite the differences in visual embedding quality across model scales. The observed performance degradation is likely due to the fact that the visual embedding quality, particularly in smaller ViTs, can negatively impact the clustering process, which in turn affects final performance. This is an interesting observation, and we plan to explore it further in future work.
> >
> > | Model | Cars | Coun | FGVC | Food | Sun | Cal | Euro | Flow | Pets | IN-1K | IN-v2 | Avg. |
> > | :--- | :---: | :---: | :---: | :---: | :---: | :---: | :---: | :---: | :---: | :---: | :---: | :---: |
> > | OpenAI CLIP ViT-B/16 | 64.78 | 22.85 | 24.21 | 88.75 | 64.36 | 82.17 | 55.91 | 71.23 | 88.93 | 68.34 | 61.85 | 63.03 |
> > | **Long-CLIP ViT-B/16** | 59.1 | 20.27 | **22.68** | 86.69 | **68.74** | 82.69 | **60.52** | **69.85** | 89.21 | **66.85** | **60.24** | **62.44** |
> > | **CLIP-FMoE ViT-B/16** | **61.15** | **22.04** | 22.2 | **87.24** | 67.87 | **82.79** | 55.74 | 69.46 | **89.56** | 65.99 | 58.92 | 62.09 |
> > | | | | | | | | | | | | | |
> > | OpenAI CLIP ViT-L/14 | 77.93 | 31.88 | 31.77 | 93.07 | 67.56 | 83.27 | 62.59 | 79.2 | 93.21 | 75.54 | 69.84 | 69.62 |
> > | **Long-CLIP ViT-L/14** | 72.37 | 28.14 | **30.78** | 91.47 | **72.54** | 82.58 | **69.09** | 72.39 | 91.03 | 72.82 | 66.88 | 68.19 |
> > | **CLIP-FMoE ViT-L-14** | **75.58** | **30.17** | 29.07 | **92.32** | 71.54 | **84.63** | 67.11 | **76.06** | **93.08** | **73.9** | **67.72** | **69.20** |
> >
> > | Model | COCO IR R@1 | COCO IR R@5 | COCO TR R@1 | COCO TR R@5 | Flickr 30k-IR R@1 | Flickr 30k IR R@5 | Flickr 30k TR R@1 | Flickr 30k TR R@5 | DCI IR R@1 | DCI IR R@5 | DCI TR R@1 | DCI TR R@5 | Avg. |
> > | :--- | :---: | :---: | :---: | :---: | :---: | :---: | :---: | :---: | :---: | :---: | :---: | :---: | :---: |
> > | OpenAI CLIP ViT-B/16 | 33.07 | 58.43 | 52.44 | 76.76 | 62.08 | 85.6 | 81.9 | 96.2 | 32.79 | 51.6 | 35.9 | 54.76 | 60.13 |
> > | **Long-CLIP ViT-B/16** | 40.92 | 66.32 | 56.94 | 80.34 | 70.38 | 90.48 | 84.1 | 98 | 46.97 | 66.66 | 41.06 | 65.47 | 67.30 |
> > | **CLIP-FMoE ViT-B/16** | **42.18** | **67.25** | **60.3** | **82.52** | **70.7** | **90.54** | **88.6** | **98.5** | **51.98** | **70.26** | **54.61** | **73.18** | **70.89** |
> > | | | | | | | | | | | | | | |
> > | OpenAI CLIP ViT-L/14 | 36.51 | 61.01 | 56.38 | 79.36 | 65.22 | 87.26 | 85.1 | 97.3 | 32.79 | 51.6 | 35.9 | 54.76 | 61.93 |
> > | **Long-CLIP ViT-L/14** | 46.97 | 71.91 | 63.32 | 84.64 | **76.08** | **93.48** | 89.3 | 98.9 | 52.67 | 70.44 | 45.32 | 70.78 | 71.98 |
> > | **CLIP-FMoE ViT-L-14** | **47.72** | **72.5** | **65.56** | **86.62** | 75.94 | 93.34 | **91.6** | **99.3** | **54.72** | **71.42** | **55.87** | **73.79** | **74.03** |

---

### Author Response · Authors · 2025-11-22
**Expressing Gratitude to Reviewers and Summary of Rebuttals**

Dear AC and Reviewers,

Thank you for your thoughtful reviews and valuable feedback, which have greatly helped us improve the paper. We are particularly encouraged by the following positive assessments:

* **Timely and relevant problem.** Reviewers noted that our work tackles an important and current challenge in *scaling CLIP with MoE under limited computational resources* and extending CLIP-style models to more diverse, realistic scenarios (Reviewers sfPq, Wka5, jDHU).

* **Clarity, organization, and rigor.** Several reviewers highlighted that the paper is *well written, clearly structured*, and that the overall framework and training pipeline are *easy to follow*, with solid motivation and technical soundness (Reviewers sfPq, Wka5, 2spG, jDHU).

* **Strong empirical support.** The experiments were praised as *comprehensive and convincing*, covering zero-shot classification, standard and fine-grained retrieval, and long-context benchmarks, with consistent improvements over CLIP-MoE and vanilla fine-tuning, while preserving the original CLIP’s zero-shot performance (average drop <1%) (Reviewers sfPq, Wka5, 2spG).

* **Computational efficiency and practicality.** Reviewers appreciated that CLIP-FMoE achieves a *favorable trade-off between performance and efficiency*, reducing training overhead compared with CLIP-MoE, keeping inference overhead moderate, and thus being *practically useful for real-world deployment* of multimodal foundation models (Reviewers sfPq, Wka5).

We also thank the reviewers for raising insightful questions and perspectives that have helped us further improve our paper. In response, we have provided additional experiments and clarifications for general concerns as follows.

* **Clarified method and positioning.** We expanded the explanation of ICCL, the Fusion Gate, and the train-before-unify design, and added dedicated related work on model merging, dynamic network composition, and continual learning methods to better position our contributions.

* **Clustering strategy and expert specialization.** We clarified the role of image-only clustering and its relation to fine-grained concepts, reported sensitivity to the number of experts and sub-clusters, provided cluster-size statistics, and added new evidence of expert specialization using the MMVP benchmark with attribute-wise analysis.

* **Scaling and generalization.** We added experiments on a larger 13.6M merged dataset with CLIP ViT-L/14 and 8 experts, comparing against smaller-scale datasets and other  OpenAI CLIP variants, and discussed how batch size and data quality jointly affect performance.

* **Expert collaboration and MoE behavior.** We conducted new studies with more experts and larger inference TopK, showing that the unified MoE model consistently outperforms the best single expert on both classification and retrieval tasks.

* **Additional ablations.** We evaluated the sensitivity of the balancing-loss coefficient α and compared freezing versus fine-tuning experts in Stage 2, showing that our default choices are empirically justified.

* **Broader applicability.** We added a discussion on how ICCL and the Fusion Gate can be extended beyond CLIP to other architectures, including LLaVA-style VLMs and more general MoE-based models.

---

### Author Response · Authors · 2025-12-02
**Briefing for the New Area Chair: Summary of Discussion Phase and Thank You for Your Consideration**

Dear new AC,

Thank you for stepping in at this challenging stage of the review process and taking over the handling of our submission. We are truly grateful for your time and support.

To summarize the ongoing discussion process, we submitted our rebuttal, and three out of four reviewers engaged in the discussion phase before the new ICLR 2026 policy was enacted. All reviewers agreed that we had adequately addressed their concerns and leaned toward acceptance; we briefly describe their main concerns and our resolution below:

* **Reviewer sfPq:** Finds the paper clear and empirically strong; the main concerns were clustering sensitivity and scalability. Our rebuttal added ablations on cluster hyperparameters, discussion of multimodal clustering, and new experiments training the model on a larger dataset (13.6M vs. 2.8M in the original results). However, due to the new policy and timeline, the reviewer did not have the opportunity to post a follow-up reply.

* **Reviewer Wka5:** Initially concerned about incremental novelty and extensibility. After our rebuttal, the reviewer posted an official comment stating that our explanations clarified the core contributions and extensibility, that the “practical recipe” for CLIP scaling and MoE adaptation is well reasoned, and that they maintain a positive score with explicit support for acceptance.

* **Reviewer 2spG**: Requested more discussion about related works on model merging, dynamic network composition, and continual learning to better highlight our contributions, as well as experiments on expert collaboration, fine-tuning experts in stage 2, and cluster distribution. After our rebuttal, the reviewer stated that our responses were sufficient and effectively resolved their previous questions, and decided to **increase the score from 4 to 6**.

* **Reviewer jDHU:** Asked for stronger evidence of expert specialization and clarification of novelty. Our new specialization analyses addressed these points; in their official comment, they state that the new experiments resolve their concerns, they have no further questions, and they keep their score marginally above the acceptance threshold, while noting that the paper is outside their main area of expertise.


We kindly ask you to consider our paper, rebuttal, and subsequent responses with additional results, bearing in mind that we have systematically addressed all points raised by the reviewers. Please feel free to contact us with any questions about the submission or our rebuttal; we would be very happy to provide further clarification or continue the discussion.

Best regards,

The Authors

---

### Meta-Review · Area_Chair_gChi · 2025-12-25

**Summary:**

This paper proposes CLIP-FMoE, combining a two-stage Isolated Constrained Contrastive Learning (ICCL) pipeline (cluster-based data partitioning to train specialized experts) with a Fusion Gate to mitigate catastrophic forgetting and preserve zero-shot capability. Reviewers’ concerns centered on (i) clustering strategy/hyperparameter sensitivity and the justification of single-modality clustering, (ii) scalability and generalization to larger data regimes, (iii) positioning/novelty of the Fusion Gate and related-work coverage, (iv) evidence of expert collaboration/synergy, and (v) ablations such as the MoE balancing-loss coefficient. The authors added clustering ablations and strategy comparisons, new experiments at a larger 13.6M data regime, expanded collaboration analysis with more experts and higher inference TopK, an ablation on α, and strengthened related work on model merging/dynamic composition/continual learning. During discussion (before the policy change), reviewers indicated the rebuttal addressed concerns and leaned toward acceptance; one reviewer increased the score from 4 to 6 and another explicitly reaffirmed support for acceptance. Overall, the work is clearly presented with broad empirical validation, and the added results resolve the main issues, so I recommend acceptance (poster).

**Reviewer Concerns:**

### Addressed by rebuttal/discussion

- **Clustering sensitivity/strategy:** sfPq raised concerns about sensitivity to clustering hyperparameters (e.g., N/M); the authors added ablations and strategy comparisons, also summarized as “cluster hyperparameters ablations / multimodal clustering discussion”.

- **Single-modality (image-only) clustering justification:** sfPq questioned whether image-only clustering captures fine-grained concepts; the authors compared multiple clustering strategies and motivated the two-level image clustering approach (also reflected in the authors’ summary).

- **Scaling/generalization:** sfPq asked about generalization beyond small-scale training; the authors added experiments on a larger 13.6M regime and discussed batch-size and data-quality effects.

- **Novelty positioning / related work:** 2spG noted the Fusion Gate novelty is incremental and requested better positioning vs model merging/dynamic composition; the authors expanded related work and positioning (also summarized by the authors).

- **Expert collaboration and α ablation:** 2spG found collaboration analysis limited and α sensitivity missing; the authors added collaboration studies with more experts and higher inference TopK and an α-sensitivity ablation suggesting robustness.

- **Extensibility beyond CLIP:** Wka5 asked about extensibility; the authors’ response was deemed sufficient, and the reviewer explicitly reaffirmed support for acceptance.


### Potentially still outstanding (not rejection-worthy, but could be strengthened)

- **No direct billion-scale training validation:** While the 13.6M setting is a meaningful step, it does not fully settle true extreme-scale behavior; authors discuss constraints and future directions.

- **Perceived incremental contribution:** Some concerns remain that the work is an engineering-focused integration of known components, although the rebuttal improved positioning and clarified the “practical recipe” contribution.

**Reviewer Scores:**

- **sfPq:** Initial rating is 6 and the reviewer indicated willingness to increase if concerns were addressed. The authors added clustering ablations and 13.6M scaling results, but the reviewer could not post a follow-up due to policy/timeline. **I expect it would stay at 6 (conservatively), with a plausible bump to 8.**

- **Wka5:** Rating 6 and explicitly reaffirmed support for acceptance. **Expected to remain 6.**

- **2spG:** The authors state the reviewer increased the score from 4 to 6 after the rebuttal. **Expected 4 → 6.**

- **jDHU:** The authors report the reviewer kept a score “marginally above the acceptance threshold,” with concerns resolved and no further questions.

---

### Decision · Program_Chairs · 2026-01-26

Accept (Poster)